# Plasmacytoid dendritic cell sensing of hepatitis E virus is shaped by both viral and host factors

Garima Joshi[1], Elodie Décembre[1], Jacques Brocard[2], Claire Montpellier[3], Martin Ferrié[3], Omran Allatif[1], Ann-Kathrin Mehnert[5], Johann Pons[6], Delphine Galiana[1], Viet Loan Dao Thi[5], Nolwenn Jouvenet[4], Laurence Cocquerel[3], Marlène Dreux[1]

**Type I and III interferons critically protect the host against viral infection. Previous studies showed that IFN responses are suppressed in cells infected by hepatitis E virus (HEV). Here, we studied the anti-HEV function of IFN secreted by plasmacytoid dendritic cells (pDCs), specialized producers of IFNs. We showed that pDCs co-cultured with HEV-replicating cells secreted IFN in a cell contact–dependent manner. This pDC response required the endosomal nucleic acid sensor TLR7 and adhesion molecules. IFNs secreted by pDCs reduced viral spread. Intriguingly, ORF2, the capsid protein of HEV, can be produced in various forms by the infected cells, and we wanted to study their role in anti-HEV immune response. During infection, a fraction of ORF2 localizes into the nucleus, and glycosylated forms of ORF2 are massively secreted by infected cells. We showed that glycosylated ORF2 potentiates the recognition of infected cells by pDCs, by regulating cell contacts. On the other hand, nuclear ORF2 triggers immune response by IRF3 activation. Together, our results suggest that pDCs may be essential to control HEV replication.**

## Introduction

Hepatitis E virus (HEV) is the most common cause of acute viral hepatitis worldwide. It has been estimated that this virus infects ~100 million people every year and is responsible for 14 million symptomatic cases and 300,000 deaths, mainly in regions of the world with poor sanitary conditions (Li et al, 2020). HEV infection is often asymptomatic and resolves on its own in case of healthy subjects. However, severe cases have primarily been reported in pregnant women, whereas chronic infections are more common in immunocompromised patients (Lhomme et al, 2020). This makes

host immunity a crucial factor in influencing the outcome of the disease. In addition, HEV infection is associated with a broad range of extrahepatic manifestations, including renal and neurological disorders (Songtanin et al, 2023). The *Paslahepevirus balayani* species of the Paslahepevirus genus contains five HEV genotypes (gt) that are pathogenic in humans. HEV gt1 and gt2 are primarily transmitted through contaminated water, exclusively infect humans, and are responsible for waterborne hepatitis outbreaks in developing countries. In contrast, industrialized countries usually fall victim to HEV gt3 and gt4, which have a zoonotic origin (Doceul et al, 2016). Chronic human infection with camel-associated HEV gt7 has also been reported (Lee et al, 2016). Alarmingly, a rat HEV from the Rocahepevirus species was recently reported to be also transmitted to humans (Sridhar et al, 2018; Andonov et al, 2019).

The most common genotype causing chronic HEV infection in the developed world is gt3 (Ma et al, 2022b). HEV infection has been recognized as a burgeoning issue in industrialized countries because of its chronicity in immunocompromised gt3-infected patients, the transmission of HEV through blood transfusion, a growing number of diagnosed HEV cases, and complications in patients with preexisting liver disease (Sayed et al, 2017). Importantly, HEV was recently ranked sixth among the top 10 zoonotic viruses presenting the greatest risk of transmission to humans (Grange et al, 2021).

HEV has a single-stranded, positive-sense RNA genome that contains three ORFs: ORF1, ORF2, and ORF3 (Tam et al, 1991). ORF1 encodes the nonstructural ORF1 polyprotein that displays domains essential for viral replication including the RNA-dependent RNA polymerase (RdRp) (Koonin et al, 1992). The RdRp produces a negative-sense RNA replicative intermediate, which serves as a template for the subgenomic RNA. The subgenomic RNA encodes both ORF2 and ORF3 proteins (Graff et al, 2006). ORF2 protein is the viral capsid protein, and ORF3 is a small multifunctional phosphoprotein, involved in particle egress (Nimgaonkar et al, 2018).

[1]CIRI, INSERM, U1111, Université Claude Bernard Lyon 1, CNRS, UMR5308, École Normale Supérieure de Lyon, University Lyon, Lyon, France   [2]Université Claude Bernard Lyon 1, CNRS UAR3444, INSERMUS8, ENS de Lyon, SFR Biosciences, Lyon, France   [3]University Lille, CNRS, INSERM, CHU Lille, Institut Pasteur de Lille, U1019-UMR 9017-CIIL-Center for Infection and Immunity of Lille, Lille, France   [4]Institut Pasteur, Université de Paris, CNRS UMR 3569, Virus sensing and signaling Unit, Paris, France   [5]Department of Infectious Diseases, Virology, Heidelberg University, Medical Faculty Heidelberg, Heidelberg, Germany and German Centre for Infection Research (DZIF), Partner Site Heidelberg, Heidelberg, Germany   [6]Sup'biotech, École Des Ingénieurs En Biotechnologies, Villejuif, Paris

Correspondence: garima.joshi@ens-lyon.fr; marlene.dreux@ens-lyon.fr
Marlène Dreux died on March 18, 2025

ORF2, which is composed of 660 amino acids, is produced in three forms: infectious ORF2 (ORF2i), glycosylated ORF2 (ORF2g), and cleaved ORF2 (ORF2c) (Montpellier et al, 2018). The precise sequences of ORF2i, ORF2g, and ORF2c proteins have been proposed by two groups (Montpellier et al, 2018; Yin et al, 2018; Ankavay et al, 2019; Hervouet et al, 2022). The ORF2i protein is not glycosylated and forms the structural component of infectious particles. In contrast, ORF2g and ORF2c proteins (herein referred to as ORF2g/c) are highly glycosylated, secreted in large amounts in the culture supernatant (i.e., about 1,000x more than ORF2i), and are the most abundant ORF2 forms detected in patient sera, where they are likely targeted by patient antibodies (Montpellier et al, 2018; Ankavay et al, 2019). Thus, ORF2g/c forms may act as a humoral decoy that inhibits antibody-mediated neutralization because of their antigenic overlap with HEV virions (Yin et al, 2018). Whether ORF2g/c proteins play a specific role in the HEV life cycle, and how ORF2 forms regulate the innate immune responses need to be elucidated. ORF2 forms are produced from different pathways, including a major one in which the ORF2 proteins are directed to the secretion pathway, where they undergo maturation and glycosylation, and are subsequently released in significant quantities. A fraction of cytosolic ORF2i proteins are delivered to the virion assembly sites (Bentaleb et al, 2022; Hervouet et al, 2022), whereas another fraction of ORF2i proteins translocate into the nucleus of infected cells, presumably to regulate host immune responses (Lenggenhager et al, 2017; Ankavay et al, 2019; Hervouet et al, 2022).

Viral genomes can be recognized by cytosolic sensors such as RIG-I, MDA5, and LGP2 (Rehwinkel & Gack, 2020), whose activation leads to the induction of type I and III IFN (IFN-I/III) pathways. HEV has developed many mechanisms to inhibit the IFN response via its three ORFs, described as follows. The HEV gt1 ORF1 protein blocks RIG-I and Tank-binding kinase 1 ubiquitination in hepatoma cells, thereby suppressing the pathway of IFN induction (Nan et al, 2014). The amino-terminal region of HEV gt3 ORF1, harboring a putative methyltransferase (Met) and a papain-like cysteine protease functional domain, inhibits IFN-stimulated response element promoter activation by inhibiting STAT1 nuclear translocation and phosphorylation (Bagdassarian et al, 2018). Moreover, HEV gt1 and gt3 ORF2 proteins antagonize IFN induction in HEV-replicating hepatocytes by inhibiting phosphorylation of the transcriptional regulator IRF3 (Lin et al, 2019). The ORF3 protein of the HEV gt3 can also suppress IFN response by blocking STAT1 phosphorylation (Dong et al, 2012).

In human hepatoma cells and primary hepatocytes, HEV infection induces only IFN-III production (Wu et al, 2018; Yin et al, 2018), which is comparatively less potent at lower doses and at early time points than IFN-I (Lazear et al, 2019). However, the elevated expression of IFN-stimulated genes (ISGs), which are effectors of IFN-I/III, was detected in the whole blood of HEV-infected patients (Yu et al, 2010; Moal et al, 2013; Sayed et al, 2017), as well as in experimentally infected mice engrafted with human hepatocytes and chimpanzee (Yu et al, 2010; Sayed et al, 2017). It must also be noted that the infectious HEV, with a complete replication cycle, has been shown to be more sensitive to IFN-α treatment than the subgenomic replicon (Zhou et al, 2016). Therefore, the host immune system can presumably mount an immune response to fight off HEV despite the attenuation of immune signaling within the host cells.

Plasmacytoid dendritic cells (pDCs), which are key producers of IFN, could be instrumental in effectively countering the evasion strategies employed by HEV within the liver microenvironment. As pDCs are resistant to virtually all viruses (Silvin et al, 2017), these do not express viral proteins that may block IFN induction. PDCs are an immune cell type known to produce up to 1,000-fold more IFNs than any other cell type (Reizis, 2019). They are thus pivotal for host control of viral infections (Webster et al, 2016; Reizis, 2019; Yun et al, 2021; Venet et al, 2023). They also recruit NK cells at the site of viral replication, favor virus-specific T-cell responses (Swiecki et al, 2010; Cervantes-Barragan et al, 2012; Webster et al, 2016, 2018; Reizis, 2019), and secrete a large panel of proinflammatory cytokines, for example, TNF-α and IL-6 (Reizis, 2019). PDC stimulation is mediated by the recognition of viral nucleic acid by TLR7 and TLR9, which localize in the endolysosomal compartment (Reizis, 2019). In the case of RNA viruses, TLR7-ligand engagement results in the formation of a signal complex comprising IRAK1 (IL-1 receptor–associated kinase 1), IRAK4, and TRAF6 (TNF receptor–associated factor 6), in a MyD88-dependent (myeloid differentiation primary-response gene 88) manner. This further triggers the translocation of IRF7 (IFN regulatory factor 7) into the nucleus, where it can induce the transcription of type I IFN genes. In addition, the NF-κB (nuclear factor-κB)– and MAPK (mitogen-activated protein kinase)-mediated signaling are also activated (Gilliet et al, 2008). It has been shown that liver pDCs retain the ability to produce copious amounts of IFN-α (Doyle et al, 2019). Whether liver-resident pDCs contribute to the control of HEV replication is unknown. Here, using hepatocyte-derived cell lines selected for their immunocompetence, we investigated the ability of HEV-replicating cells to stimulate pDCs, and defined how ORF2 forms contribute to this process.

## Results

### Immunocompetence of HEV cellular models

The magnitude of the type I and III IFN (herein referred to as IFN-I/III) responses against viruses varies among cell types. We thus first assessed the immunological robustness of three cell lines, namely, Huh-7.5, HepG2/C3A, and PLC3 cells, which are permissive to HEV (Shukla et al, 2011; Schemmerer et al, 2016; Montpellier et al, 2018). Huh 7.5 cells are a subclone of the hepatoma Huh-7 cells that express an inactivate version of RIG-I and thus display an increased permissiveness to viral infection (Blight et al, 2002; Sumpter et al, 2005). HepG2/C3A cells resemble liver parenchymal cells and were selected from the primary hepatoblastoma–derived cell line HepG2 for cell contact inhibition, leading to a more hepatocyte-like phenotype compared with the parental line (Knowles et al, 1980). PLC3 cells are a subclone of the PLC/PRF/5 hepatoma cell line broadly used to study HEV (Montpellier et al, 2018). We measured mRNA levels of two IFN-stimulated genes (*MXA* and *ISG15*) and two cytokines (*TNF* and *IL-6*) in the three cell lines upon stimulation with polyI:C (Fig 1A–C), which is a synthetic analog of dsRNA that activates TLR3, when added to the cell culture, and it activates the cytosolic sensors RIG-I and MDA5 when transfected into cells. *MXA* and *ISG15* are representatives of the IFN-I/III responses induced via IRF3-

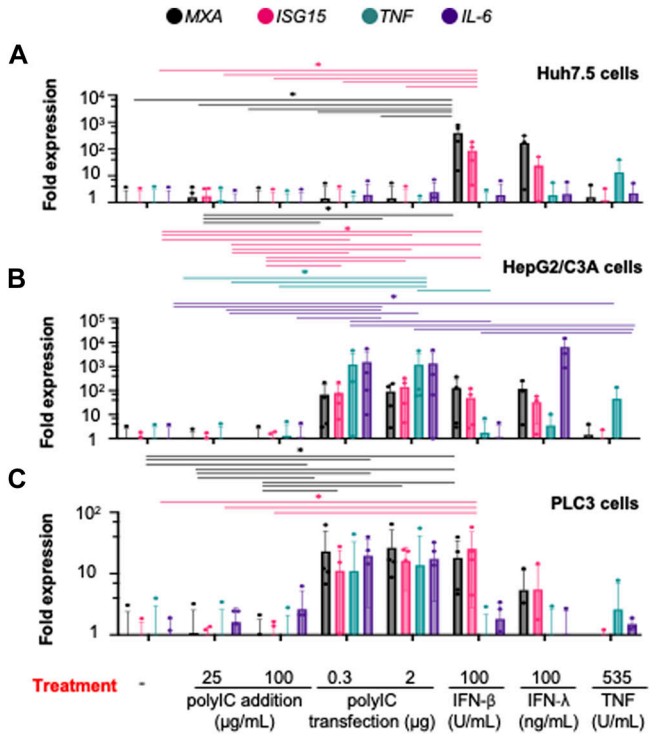

**Figure 1. Immunoresponsiveness of cell types selected as highly susceptible to HEV infection.**
**(A, B, C)** Quantification of transcript expression levels of representatives of the IFN-I/III–related pathway, that is, *MXA* and *ISG15*, NF-KB–induced signaling, that is, *TNF* and *IL-6*, upon stimulation by agonists of TLR3, that is, addition of polyI:C; RIG-I/MDA-5 cytosolic sensors by transfection of polyI:C LMW and addition of recombinant IFN-β, IFN-λ3, and TNF at the indicated concentrations for 6-hour incubation, determined by RT–qPCR. **(A, B, C)** Analyses were performed in Huh-7.5 cells (A), HepG2/C3A cells (B), and PLC3 cells (C); bars represent fold expression; means ± SD; each dot represents one independent experiment, n = 4 for treatments including polyI:C addition/transfection and IFN-β treatment, n = 3 for treatment with IFN-λ and TNF; statistical analysis was performed using the paired pairwise Wilcoxon test; *P*-values: * ≤ 0.05, ** ≤ 0.005, and *** ≤ 0.0005.

mediated signaling, whereas *TNF* and *IL-6* are induced by NF-κB signaling. All three cell types were deficient in TLR3 activity as none of them responded to treatment with polyI:C (Fig 1A–C). The expression of *MXA*, *ISG15*, *TNF*, and *IL-6* remained unaffected upon polyI:C transfection in Huh-7.5 cells (Fig 1A). In contrast, the expression of these four genes was up-regulated upon activation of RIG-I and MDA5 in both HepG2/C3A and PLC3 cells (Fig 1B and C). Treatment with different IFN types and TNF-α has been shown to inhibit HEV replication to a certain extent (Todt et al, 2016; Wang et al, 2016; Zhou et al, 2016; Murata et al, 2020). Consequently, we determined whether the three cell lines responded to stimulation by three recombinant cytokines: IFN-β, IFN-λ3 (i.e., representatives of type I and III IFNs, respectively), or TNF-α. We found that IFN-β treatment induced the expression of *MXA* and *ISG15* in all three cell lines but barely induced the proinflammatory cytokines, *TNF* and *IL-6* (Fig 1A–C). The three cell lines responded to the treatment with recombinant IFN-λ with an up-regulation of *MXA* and *ISG15* mRNA (Fig 1A–C). However, only HepG2/C3A cells showed significant

induction of *IL-6* upon IFN-λ treatment (Fig 1B). Treatment with TNF-α mainly up-regulated the proinflammatory cytokine *TNF* in the three cell lines (Fig 1A–C). Together, our results show that HepG2/C3A and PLC3 cells are sufficiently immunocompetent and were therefore selected for further investigation.

Next, PLC3 and HepG2/C3A cells were electroporated with capped RNA of the HEV gt3 p6 strain (Shukla et al, 2012). RT–qPCR analyses showed that HepG2/C3A and PLC3 cells produced a significant amount of viral RNA (Fig S1A and B) compared with mock-electroporated cells (control). Immunostaining of the ORF2 protein in PLC3 and HepG2/C3A cells at 6 days post-electroporation (d.p.e.) confirmed viral replication (Fig S1C). In addition, we optimized a staining protocol to quantify ORF2-positive cells by flow cytometry. Flow cytometry analyses showed that about 50% of PLC3 cells were positive for ORF2 at 6 d.p.e. (Fig S1D). Overall, these results indicated that HepG2/C3A and PLC3 cells are suitable models to perform immunological studies in the context of HEV replication.

### Activation of pDC upon contact with HEV-replicating cells leads to antiviral response

Next, we looked at expression levels of three ISGs (*MXA*, *ISG15*, and *OAS2*) and three cytokines (*TNF*, *IL-6*, and *IFN-λ1*) in HEV-replicating HepG2/C3A and PLC3 cells (Fig 2A and B). We found a significant increase in mRNA levels of *ISG15*, *IL-6*, and *IFN-λ1* in HepG2/C3A cells at 6 d.p.e. (Fig 2A; left panel). *OAS2*, whose encoded protein is known to activate RNase L antiviral activity, was also induced in HepG2/C3A (albeit not significantly). In PLC3 cells, mRNA abundance of *MXA*, *ISG15*, *IL-6*, *OAS2*, and *IFN-λ1* remained unchanged (Fig 2B; left panel). These results are in accordance with the higher responsiveness of HepG2/C3A cells to activation of cytosolic sensors and cytokines as compared to PLC3 cells (Fig 1B and C). Notably, the level of *IFN-λ1* mRNA, which has a known antiviral role against HEV in mice (Sari et al, 2021), increased upon viral replication only in HepG2/C3A cells (Fig 2A and B; left panels). In addition, *TNF* expression was slightly inhibited in HEV-replicating PLC3 cells (Fig 2B; left panel) in accordance with previous results (Hervouet et al, 2022). These results showed that although HepG2/C3A and PLC3 cells are immunocompetent upon stimulation (Fig 1), IFN-I/III response is only modestly induced upon HEV replication in HepG2/C3A cells and marginally in PLC3 cells. In line with previous in vitro studies (Dong et al, 2012; Nan et al, 2014; Bagdassarian et al, 2018; Lin et al, 2019), these results suggest that HEV is a poor inducer of IFN response.

In vivo studies of HEV infection (i.e., at the tissue level) have, nonetheless, demonstrated induction of antiviral responses (Yu et al, 2010; Moal et al, 2013; Sayed et al, 2017). We thus thought to investigate whether human primary pDCs can mount the IFN-I/III response. The human primary pDCs were stimulated upon co-culture for 18 h with HEV-replicating cells. As in aforementioned experiments, we analyzed mRNA abundance of *MXA*, *ISG15*, *IFN-λ1*, *OAS2*, *TNF*, and *IL-6* by RT–qPCR (Fig 2A and B; right panels). We observed up-regulated expression of all these effectors, and among these, *MXA*, *ISG15*, *IFN-λ1*, and *OAS2* were significantly up-regulated when pDCs were co-cultured with HEV-replicating HepG2/C3A cells, as compared to basal levels with uninfected cells (Fig 2A and B; right panels). Overall, likely because of the inhibitory mechanisms of HEV

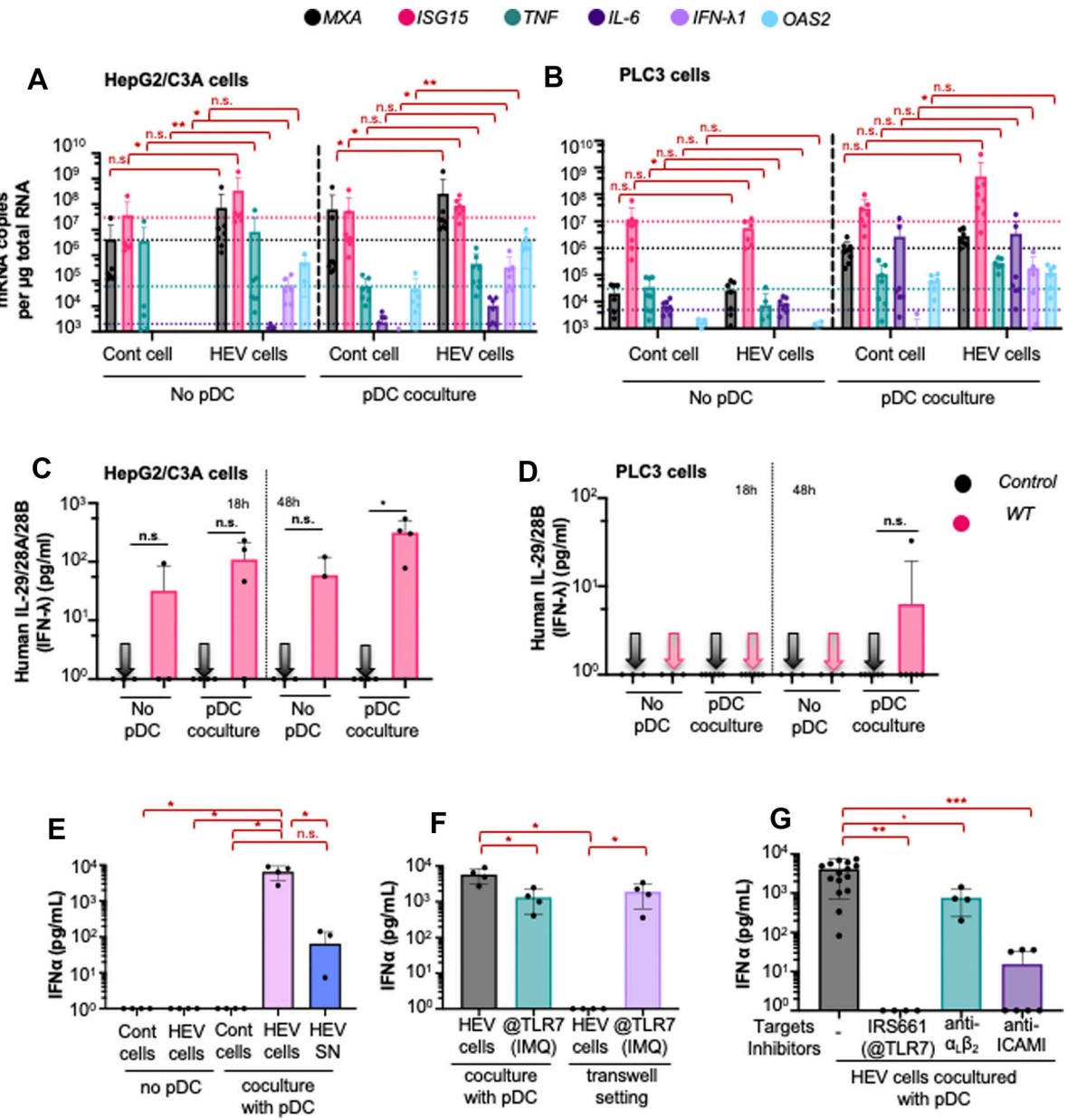

**Figure 2. pDC response against HEV-infected cells.**

**(A, B)** Quantification of the transcript expression levels of representatives of IFN-I/III signaling (i.e., *MXA*, *ISG15*, *OAS2*, and *IFN-λ1*) and NF-KB–related pathway (i.e., *TNF* and *IL-6*) determined at 6 d.p.e. of HepG2/C3A (A) and PLC3 (B) cells that were electroporated with 10 μg RNA [HEV cells] or mock-electroporated without RNA [cont cells], in the absence (left panels) or in co-culture with pDCs for 18 h (right panels), as indicated. No pDC and pDC co-culture conditions have been separated into left and right panels as steady-state levels of gene expression are different in these two datasets, and therefore, cross-comparisons between the two panels must be avoided. Bars represent copy number per μg total RNA; means ± SD; each dot represents one independent experiment (n = 3–8). Statistical analysis was done using the Wilcoxon rank-sum test with continuity correction; *P*-values: * ≤ 0.05, ** ≤ 0.005, and *** ≤ 0.0005. **(C, D)** pDCs were co-cultured with HEV-electroporated PLC3 or HepG2/C3A cells for 18 and 48 h. **(C, D)** Quantification of IL-29/28A/28B in supernatants of pDCs co-cultured with HEV-replicating HepG2/C3A (C) and PLC3 cells (D); n ≥ 3; statistical analysis was done using the Wilcoxon rank-sum test with continuity correction; *P*-values: * ≤ 0.05, ** ≤ 0.005, and *** ≤ 0.0005. **(E, F, G)** pDCs were co-cultured with HEV-electroporated PLC3 cells, and their supernatants, in various settings, or treated by inhibitors, as indicated, for 18 h. **(E)** Quantification of IFN-α in supernatants of pDCs co-cultured with HEV-replicating PLC3 cells [HEV cells] or treated with supernatants from HEV-infected cells [HEV SN] versus in the absence of pDCs [no pDC]; the uninfected [cont] cells were electroporated without HEV RNA and used as a negative control. Bars represent means ± SD, and each dot represents one independent experiment (n = 4). **(F)** Quantification of IFN-α in supernatants of pDCs in co-culture or separated from HEV-electroporated PLC3 cells by a permeable membrane (0.4 μm) of transwell [transwell setting]. The TLR7 agonist, imiquimod [IMQ], was used as a positive control. **(E)** Results presented as in (E); n = 4. **(G)** Co-culture of pDCs and HEV-electroporated PLC3 cells was treated by inhibitors of TLR7 [IRS661], blocking antibodies against $\alpha_L\beta_2$-integrin and ICAM-1, followed by the quantification of IFN-α in supernatants of the co-cultures. **(E)** Results presented as in (E); n > 3 independent experiments. Statistical analysis was done using the Wilcoxon rank-sum test with continuity correction; *P*-values: * ≤ 0.05, ** ≤ 0.005, and *** ≤ 0.0005.

viral products in PLC3 cells targeting IFN-I/III amplification pathways, there was a poor activation of downstream effectors, ISGs, and cytokines in the mixed cell culture, with the exception of *IFN-λ1*. The induction of *IFN-λ1* was common to the pDC co-cultures of HEV-replicating HepG2/C3A and PLC3 cells. We tested whether this was true at the protein level by ELISA for lambda IFNs or type III IFNs (IFN-λ1 [IL-29], IFN-λ2 [IL-28A], and IFN-λ3 [IL-28B]). We observed that IFN-III protein levels were significantly up-regulated only at 48 h post-pDC co-culture with HEV-replicating HepG2/C3A cells, as compared to control cells co-cultured with pDCs (Fig 2C). Even though *IFN-λ1* mRNA levels were up-regulated upon co-culture of pDCs with HEV-replicating PLC3 cells as early as 18 h of co-culture (Fig 2B; right panel), only a small amount of secreted IFN-III was detectable at the protein level 48 h post-co-culture (Fig 2D). This is possibly because of the lesser sensitivity of the ELISA over RT–qPCR analyses (i.e., lower dynamic range). Further analyses with more sensitive/advanced approaches for IFN-III detection will be needed. Therefore, IFN-III is more robustly up-regulated in pDC co-culture of HEV-replicating HepG2/C3A cells.

Next, we tested whether pDCs can produce IFNs in response to HEV-replicating cells by quantifying the secreted levels of multiple subtypes of IFN-α as representative of IFN-I signaling (Fig 2E). HEV-replicating PLC3 cells alone did not produce detectable IFN-α levels, in accordance with previous findings, showing that HEV replication does not induce IFN-I secretion (Yu et al, 2010; Moal et al, 2013; Nan et al, 2014; Sayed et al, 2017). In contrast, when pDCs were co-cultured with HEV-replicating PLC3 cells, more than 1,000 pg/ml of IFN-α was secreted, whereas no IFN-α was detected when pDCs were co-cultured with uninfected control cells (Fig 2E). We also treated pDCs with cell-free HEV to verify whether circulating virus particles can also activate pDCs in a cell-independent manner. PDCs exposed to the filtered supernatant [SN] collected from HEV-replicating cells very modestly secreted IFN-α (Fig 2E), suggesting that physical contact between pDCs and HEV-replicating cells is required for the robust pDC IFN-I secretion. Therefore, we assessed whether HEV-replicating PLC3 cells activated pDCs when the two cell types were separated by a 0.4-μm permeable membrane, which allows diffusion of virions but not cells. In this transwell setting, pDCs placed in the top chamber did not produce any IFN-α, validating that physical contact between the cells is required for pDC stimulation (Fig 2F). Imiquimod [IMQ], which is a soluble agonist of TLR7 (Gibson et al, 2002), was used as an additional positive control to rule out nonspecific effect of the setting (Fig 2F). Furthermore, when the co-cultures were treated with blocking antibodies against ICAM-I and $\alpha_L\beta_2$-integrin, two cell–cell adhesion proteins that mediate cellular contacts (Marlin & Springer, 1987; Assil et al, 2019b), pDC response to infected cells was significantly reduced as compared to untreated pDCs in co-culture with HEV-replicating cells (Fig 2G). Because TLR7 is responsible for sensing replicative viral RNAs in pDCs (Webster et al, 2016; Reizis, 2019; Assil et al, 2019b), we tested its contribution in pDC response to HEV-replicating cells. When the co-cultures were treated with the TLR7 inhibitor IRS661, no IFN-α was secreted (Fig 2G), suggesting that stimulation of pDCs by HEV-replicating cells is mainly mediated by TLR7. Together, our results show that direct contacts mediated by ICAM-I and, at least in part, with $\alpha_L\beta_2$-integrin, enable pDCs to sense and respond to HEV-replicating cells in a TLR7-dependent manner.

We then tested whether IFN-α produced by pDCs inhibited viral propagation in either HepG2/C3A or PLC3 cells (Fig 3A–C). Upon 18 h of co-culture of pDCs with HEV-replicating HepG2/C3A cells, we observed a 1-log decrease in viral RNA yield (readout for viral replication) compared with the absence of pDCs (Fig 3A). At this early time of co-culture, the impact of pDCs on viral RNA replication was not yet observed for PLC3 cells (Fig 3B). This may result from a stronger up-regulation of ISGs in HEV-replicating HepG2/C3A cells (in the absence of pDCs) compared with PLC3 cells, which may result in the synergy of the antiviral impact of pDCs, leading to an earlier decrease in viral replication in HepG2/C3A cells. As we did not observe pDC-mediated viral control in HEV-replicating PLC3 cells at this early time point, we thus studied the effect of pDCs when co-cultured for 48 h. After a longer co-culture of 48 h with pDCs, we observed a 50% reduction of HEV-replicating cells as compared to cultures without pDCs (Fig 3C).

To further examine whether the pDC-mediated IFN response decreases the percentage of newly infected cells, we co-cultured a mixture of PLC3 cells, electroporated with HEV RNA and not expressing GFP (ORF2+/GFP– cells) along with uninfected cells positive for GFP (ORF2–/GFP+) in the presence or absence of pDCs for 48 h. The impact of pDC response on HEV spread was assessed by flow cytometry to quantify the percentage of newly infected cells, that is, GFP+ cells that became ORF2+ because of viral spread (ORF2+/GFP+) in the presence versus absence of pDCs (Fig 3D and E). Infected cells were distinguished from pDCs on the basis of size (FSC-SSC gating), and then, the infected cell type (PLC3 cells) was gated for the expression of ORF2 and/or GFP (Fig 3D). The results demonstrated that pDC response reduced viral replication in cells replicating HEV (Fig 3E, blue bars) and controlled viral spread to naive GFP+ cells by half, upon prolonged co-culture (Fig 3E, green bars). Collectively, our results showed an effective control of HEV spread by pDC-mediated antiviral activities involving IFN-I production.

## Modulation of pDC response by HEV-expressing ORF2 mutant is dependent on the host cell type

PDCs responded to contact with HEV-replicating cells via TLR7 recognition (Fig 2E), suggesting that the HEV genome is transferred from infected cells to the endosomal compartments of pDCs, where TLR7 localizes (Dreux et al, 2012). The distinctive feature of the HEV ORF2 capsid protein is its generation in three distinct forms (ORF2i, ORF2g, and ORF2c), each with different localization and trafficking patterns within infected cells (Lenggenhager et al, 2017; Ankavay et al, 2019; Hervouet et al, 2022). To address whether these distinct ORF2 forms impact the host immune responses in infected cells, and consequently and/or in addition the sensing of infected cells by pDCs, we tested a series of ORF2 protein mutants of the p6 HEV strain (WT) (Fig S1E). The first mutant, called 5R/5A mutant, expresses an ORF2 in which the arginine-rich motif located in the ORF2 N-terminal region, serving as a nuclear localization signal, was mutated and thus prevents ORF2 translocation into the nucleus (Hervouet et al, 2022). The nuclear export signal (NES) mutant expresses an ORF2 that lacks one of its NES, leading to its retention in the nucleus (Hervouet et al, 2022). Glycosylated proteins can influence immunological functions via their

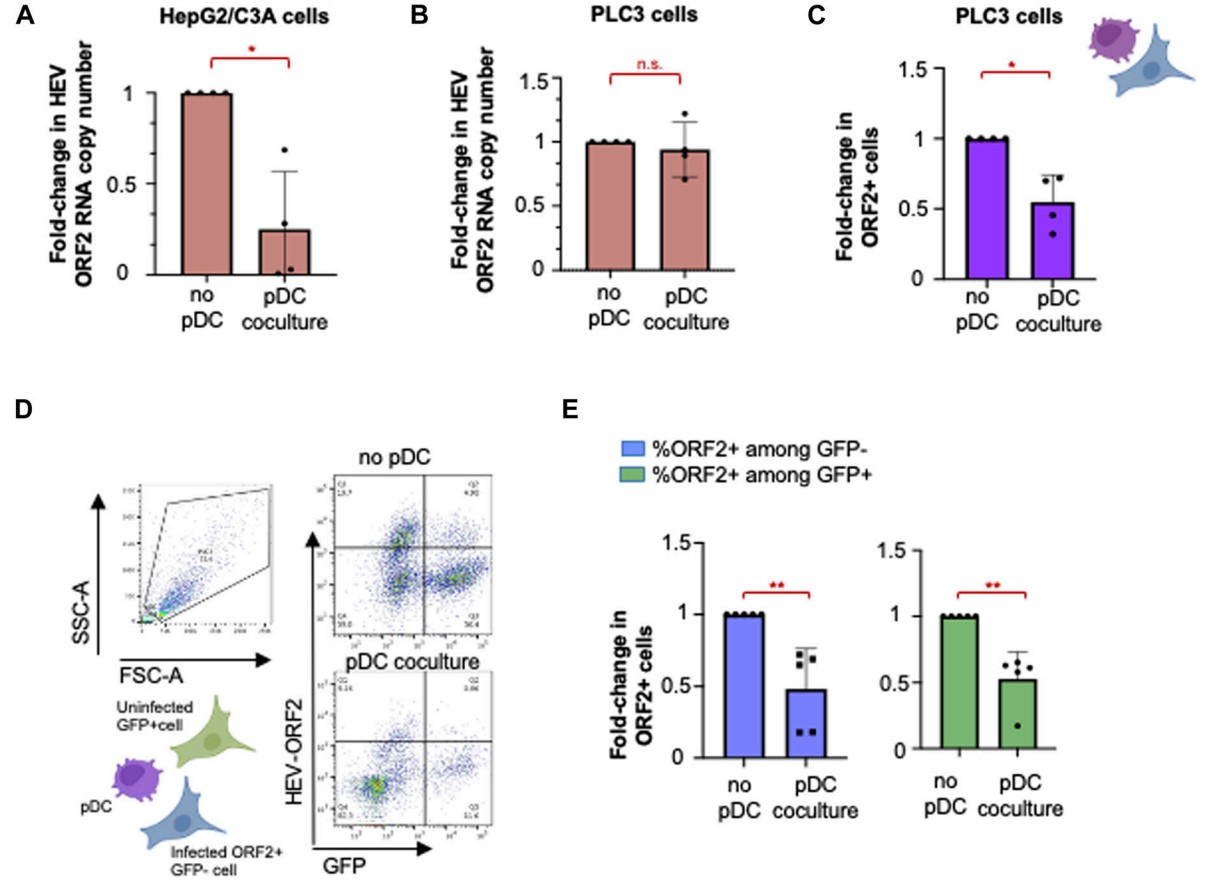

**Figure 3. Control of HEV infection by pDCs.**
**(A, B)** Quantification of HEV RNA replication levels in HepG2/C3A (A) and PLC3(B) cells, in the absence or in co-culture with pDCs, as indicated; means ± SD; each dot represents one independent experiment in terms of fold expression compared with [no pDC] condition; n = 4 for HepG2/C3A cells and PLC3 cells. Statistical analysis was done using the Wilcoxon rank-sum test with continuity correction and *P*-value adjustment with the Bonferroni method; *P*-values: * ≤ 0.05, ** ≤ 0.005, and *** ≤ 0.0005. **(C)** Quantification of ORF2-expressing cells in the presence or absence of pDCs by flow cytometry; fold change in HEV-replicating (ORF2+) PLC3 cells in the absence and presence of pDCs for 48 h (n = 4). **(D, E)** HEV-replicating PLC3 cells (GFP– ORF2+) and uninfected cells (GFP+ ORF2–) were co-cultured in the presence and absence of pDCs for 48 h. **(D)** Quantification of ORF2- and GFP-expressing cells by flow cytometry; (D) results are presented as representative dot blots. **(E)** Fold change in ORF2+ GFP– and ORF2+ GFP+ cells was quantified by flow cytometry. Bars represent means ± SD, and each dot represents one independent experiment (n = 5). Statistical analysis was done using the Wilcoxon rank-sum test with continuity correction and *P*-value adjustment with the Bonferroni method; *P*-values: * ≤ 0.05, ** ≤ 0.005, and *** ≤ 0.0005.

recognition by C-type lectin receptor (Cambi et al, 2005), either by weakening TLR7- and TLR9-mediated IFN-I/III response (Meyer-Wentrup et al, 2008; Florentin et al, 2012) or by contributing to cell interaction and viral uptake (Bermejo-Jambrina et al, 2018). Therefore, we generated an additional mutant, called STOP mutant, expressing an ORF2 protein with a nonfunctional signal peptide, preventing its translocation into the endoplasmic reticulum and thus production of glycosylated ORF2g/c forms, as recently described (Nagashima et al, 2023). This mutant carries a stop codon in the ORF2 signal peptide, which does not affect ORF3 expression. In the STOP mutant, ORF2 protein translation starts at the first initiation codon (Met1), stops at the stop codon (*10), and restarts at the second initiation codon (Met16) (Fig S2A). PLC3 cells expressing this mutant produce intracellular ORF2i form (Fig S2B; right panel), efficiently replicate HEV genome (Fig S2D), and produce HEV particles (Fig S2C, IP P1H1, and Fig S2E and F), but no ORF2g/c proteins (Fig S2C, IP P3H2). It is noteworthy that we observed the absence of ORF2g/c production in cell culture

supernatants of the STOP mutant expressed in both PLC3 and HepG2/C3A cells (Fig S2G). Finally, a previously characterized replication-defective p6 mutant (GAD) was also included in the analysis (Emerson et al, 2013).

The subcellular distribution of ORF2 mutants, compared with WT ORF2, was first characterized using confocal microscopic analyses. The WT ORF2 was found in both cytoplasm and nucleus, in both HepG2/C3A and PLC3 cells (Fig 4A and B). The 5R/5A mutant was excluded from the nucleus, whereas the NES mutant accumulated in the nucleus in both cell types (Fig 4A and B), in accordance with previous observations (Ankavay et al, 2019; Hervouet et al, 2022). The STOP mutant exhibited localization similar to that of WT ORF2 (Fig 4A and B). As expected, the GAD mutant, in which the polymerase active site was mutated and thus is replication-defective, did not produce ORF2 (Fig 4A and B). We then assessed viral RNA yield produced in cells electroporated with the four mutant genomes or with the WT genome using RT–qPCR analysis. This attested comparable levels of HEV RNA, which is a prerequisite for testing the

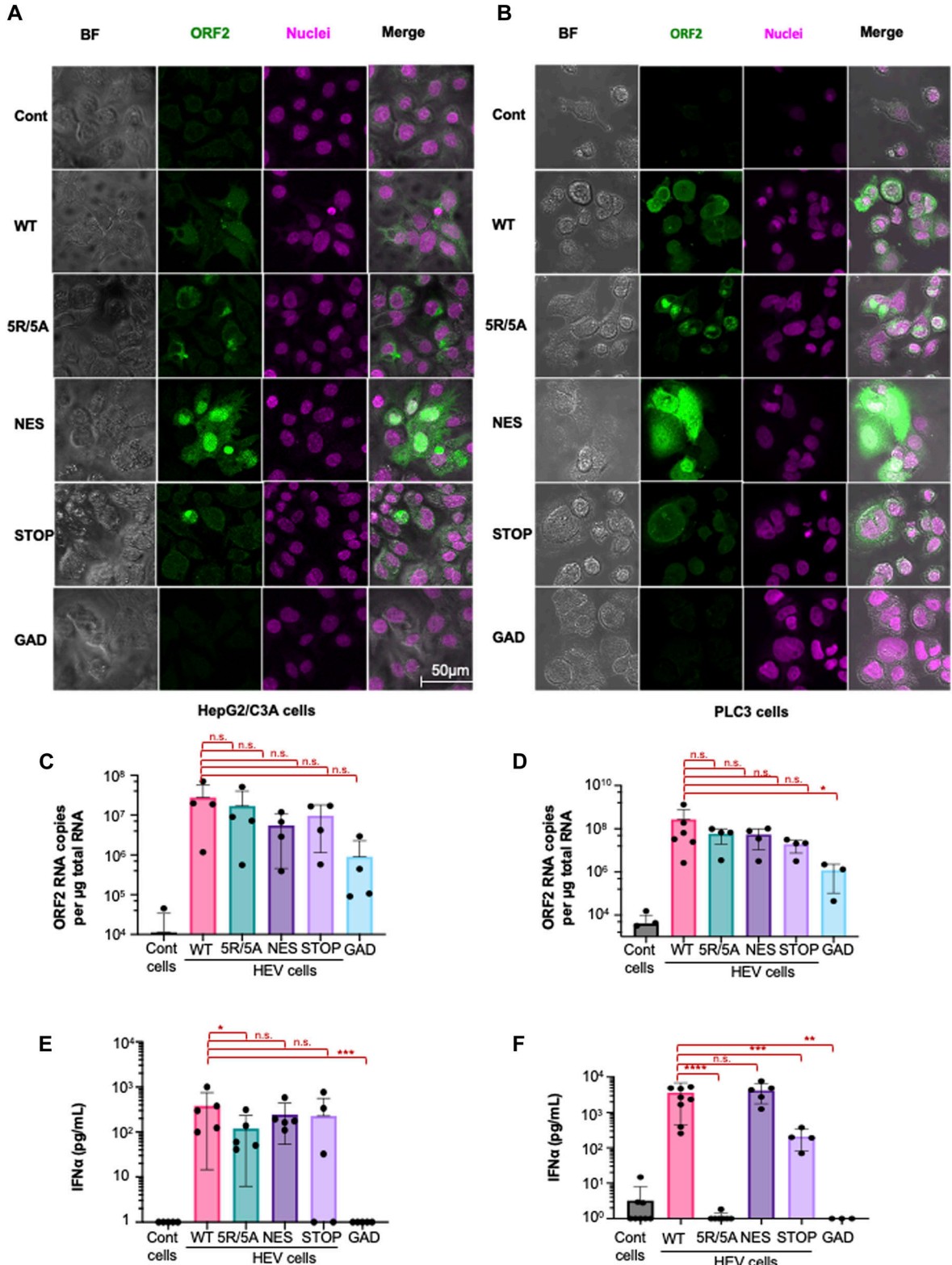

**Figure 4. Impact of ORF2 nuclear localization on pDC activation by studying co-cultures with specific HEV mutants.**
Mutations were designed to inactivate i) nuclear translocation of ORF2 as [5R5A] mutant, ii) the ORF2 export from nucleus as [NES] mutant, and iii) ORF2g/c secretion as [STOP] mutant. Cells harboring the [GAD] mutant that is deficient for viral replication and/or uninfected [cont] cells were used as negative controls. Plasmids containing the above-mentioned mutations in the HEV genome versus WT served as templates for in vitro transcription into HEV RNA, then transfected in either HepG2/C3A or PLC3 cells, as indicated. **(A, B)** Intracellular distribution of ORF2 (1E6 antibody, followed by targeting secondary antibody with Alexa Fluor 488) in HepG2/C3A (A) or PLC3 (B)

impact of this mutant panel on the response of co-cultured pDCs. The HEV RNA levels were similar across the mutants and WT genomes at 6 d.p.e. in the HepG2/C3A cells (Fig 4C). PLC3 cells electroporated with the WT, 5R/5A, NES, and STOP mutant genomes also yielded similar levels of viral RNAs at 6 d.p.e. (Fig 4D). About two-log less viral RNA was recovered in HepG2/C3A and PLC3 cells expressing the GAD genome mutant (Fig 4C and D). Viral RNA detected in these cells likely represents "input" RNA, that is, electroporated viral RNA.

Quantification of IFN-α production by pDCs co-cultured with HepG2/C3A expressing the different mutants demonstrated significant differences. First, pDCs co-cultured with HepG2/C3A cells expressing the GAD mutant did not trigger a pDC response (Fig 4E), nor in the context of PLC3 cells (Fig 4F). Because HEV RNA levels were reduced for the GAD mutant, it is tempting to hypothesize that active viral replication is required for mounting a pDC-mediated viral response. Next, pDCs co-cultured with HepG2/C3A expressing 5R/5A demonstrated a significant reduction in IFN-α production as compared to WT, and 5R/5A versus NES (Fig 4E). These results suggested that in this cellular model, the nuclear localization of ORF2 influences pDC stimulation. Along the same lines, pDC-mediated IFN-α response was completely abolished upon co-culture with PLC3 cells expressing the 5R/5A mutant (Fig 4F). Because comparable HEV RNA levels were quantified in NES versus 5R/5A mutant (Fig 4D), the nuclear localization of ORF2, which is abrogated for the 5R/5A mutant, may affect pDC response against HEV. Because the 5R/5A mutant also does not produce infectious viruses (Hervouet et al, 2022), pDC response against HEV might be modulated by viral particle release and/or nuclear ORF2.

IFN-α production was lower when pDCs were co-cultured with PLC3 cells expressing the STOP mutant, as compared to cells expressing WT ORF2, suggesting that ORF2g/c forms contribute to the pDC IFN-I response in these cells. The ORF2g/c forms produced by the WT ORF2 may be recognized by the C-type lectin receptors of pDCs, contributing to an enhanced response to immunostimulatory RNA. Alternatively, ORF2g/c expression may facilitate viral RNA transfer to pDCs and its subsequent recognition by TLR7 in endosomes. However, the decrease in pDC response to the STOP mutant was not observed for HepG2/C3A cells (Fig 4E).

Here, we showed that the pDCs exhibited reduced IFN-α response to HepG2/C3A expressing the 5R/5A mutant, and a complete absence of this response in PLC3 cells. This is likely because of differences in immune signaling among the two cell lines with the 5R/5A mutant. We observed that HEV infection impacted, in a cell type–dependent manner, the expression of *TNF* (Fig 2A), a representative cytokine of NF-κB signaling, known to modulate the expression of regulators of pDC adhesion and/or recruitment, including ICAM-I (Kim et al, 2008) and various inflammatory chemokines (Sedger & McDermott, 2014). We thus decided to analyze the effect of 5R/5A mutation

on *TNF* expression in these two cell lines (Fig S3A and B). We found that the 5R/5A mutant significantly reduced *TNF* induction in PLC3 cells but not in HepG2/C3A cells as compared to HEV WT, thus potentially explaining the cell type–specific regulation by ORF2i. Taken together, our results show that ORF2 protein expression and localization in producer cells modulate pDC response, with a magnitude that is cell type–dependent.

## ORF2 protein forms contribute to the robustness of contact between replicating cells and pDCs

We sought to determine whether HEV ORF2-mediated regulation of the strength and duration of cell-to-cell contacts could impact pDC response to infected cells. To achieve this, we developed a confocal imaging pipeline that quantifies cell proximity between pDCs and infected cells (Fig S4A). Because more contrasting effects among ORF2 mutants were observed in PLC3 cells than in HepG2/C3A cells (Fig 4), we selected PLC3 cells for investigating the role of cell-to-cell contacts in pDC-mediated response. The pDCs and HEV-replicating cells were stained using CellTrace Violet (CTV) and CellTracker Red (CMTPX), respectively, and then co-cultured and fixed after 4 h (Fig 5A). HEV-replicating cells were identified using the P3H2 anti-ORF2 antibody, which recognizes all forms of ORF2 (ORF2g/c/i) (Bentaleb et al, 2022). The raw images were analyzed by an ImageJ macro-driven automatic analysis of cell–cell contacts. The two criteria for selection of contacts between pDCs and infected cells were the distance between the two cell types (1 μm), and a contact area (>0 μm$^2$) between the surfaces of the two cell types (Fig S4B).

The percentage of ORF2-positive cells was within a comparable range among WT and the mutants in PLC3 cells (Fig 5B), in agreement with their similar RNA levels (Fig 4D). As compared to co-culture of pDCs with cells expressing WT ORF2, fewer cell-to-cell contacts (Fig 5C) were observed between pDCs and cells expressing the 5R/5A (i.e., reproducible but not significant), and a significant 40% decline in pDC contacts with cells expressing the STOP mutant. Taking into consideration that the STOP mutant displayed a reduced induction of pDC response (Fig 4F), this decline in pDC response to the STOP mutant could be due to reduced cell-to-cell contacts. Nevertheless, no difference was observed in the quality or strength of contacts, expressed in terms of contact area, across the mutants and WT (Fig S4C). This indicates that pDC function is likely influenced by frequency of contact formation rather than the area of contact between the infected cell and the pDC, which was found to be consistent across the WT and mutants.

Next, to further investigate the effect of ORF2 nuclear localization on immune signaling, we carried out single-cell imaging flow cytometry experiments in HepG2/C3A cells (Fig 6A), which had more robustly up-regulated ISGs upon HEV infection independently of

---

cells electroporated with different mutants of HEV. **(C, D)** Quantification of HEV replication levels of mutants versus WT in HepG2/C3A (n = 4) (C) or PLC3 cells (n = 3–6) (D). Results represent HEV RNA copy number (primer/amplicon designed in ORF2 RNA) per μg total RNA, detected by RT–qPCR; bars represent means ± SD, and dot represents independent experiments. **(E, F)** Quantification of IFN-α in supernatants of pDCs that were co-cultured with cells harboring HEV with the indicated mutant genome versus WT, and GAD or uninfected, as negative controls; for 18 h; 6 d.p.e. of HepG2/C3A cells (n = 5) (E) and PLC3 cells (n = 3–8) (F); bars represent means ± SD, and dots represent independent experiments. Statistical analysis was done using the Wilcoxon rank-sum test; *P*-values: * ≤ 0.05, ** ≤ 0.005, and *** ≤ 0.0005.

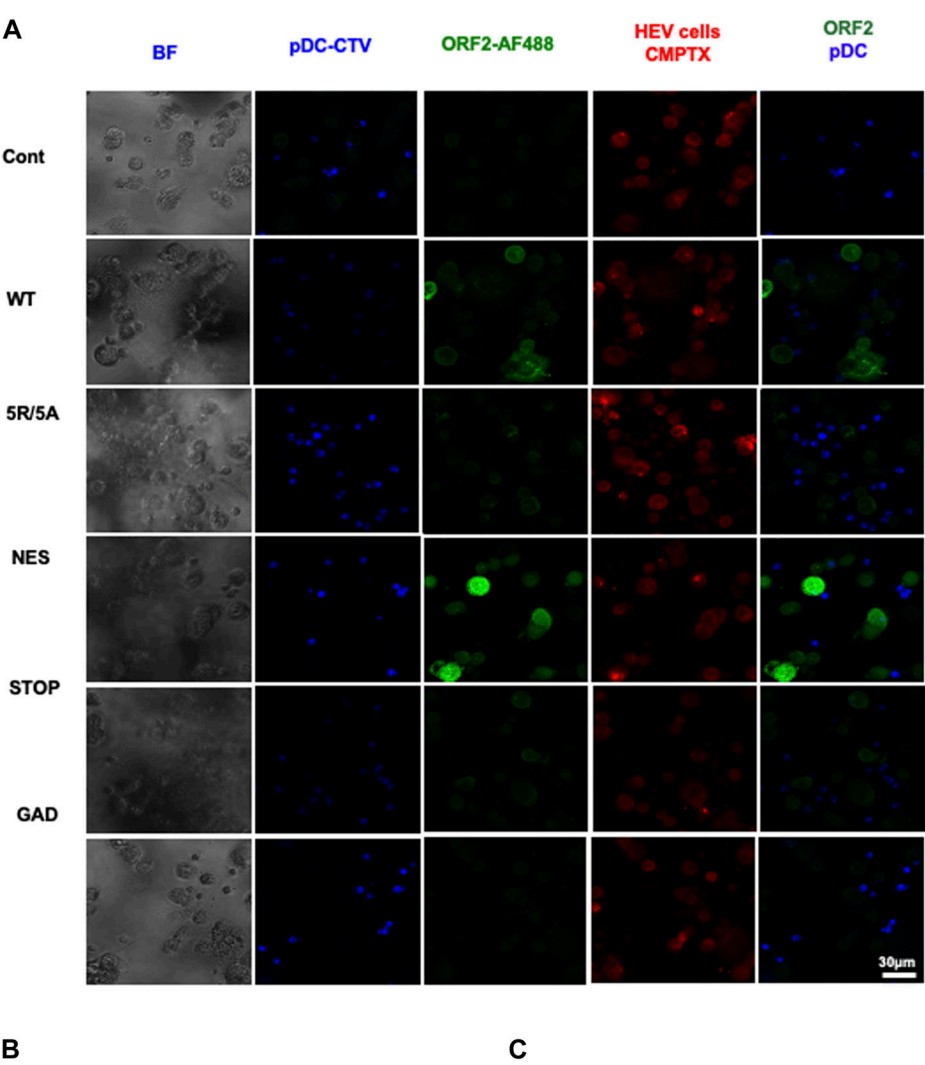

Figure 5. Effect of ORF2 expression on localization on physical contacts between HEV cells and pDCs.
(A, B, C) pDCs co-cultured with HEV-replicating PLC3 cells 6 d.p.e, bearing the indicated mutations versus WT and GAD, as reference and negative control, respectively. Confocal imaging of pDCs and HEV co-culture performed after 4-h incubation. (A) Representative confocal stack imaging of ORF2 immunostaining (green) in infected PLC3 cells, which were stained with CellTracker Red before co-cultures (CMTPX, red), combined with pDCs stained with Cell-Tracer Violet before co-culture (CTV; blue). (B) Frequency of HEV-replicating (ORF2+) PLC3 cells detected among the cells defined as non-pDC (CTV–/CMTPX+); bars present means ± SD and each dot for independent image (n ≥ 8) from four distinct experiments. (C) Contact/proximity of PLC3-infected cells and pDCs with detection of CTV/pDCs and CMTPX+ ORF2+/infected cells, as reference; bars present means ± SD and each dot for independent image (n ≥ 8) from four distinct experiments. Statistical analysis was done using the Wilcoxon rank-sum test; P-values: * ≤ 0.05, ** ≤ 0.005, and *** ≤ 0.0005.

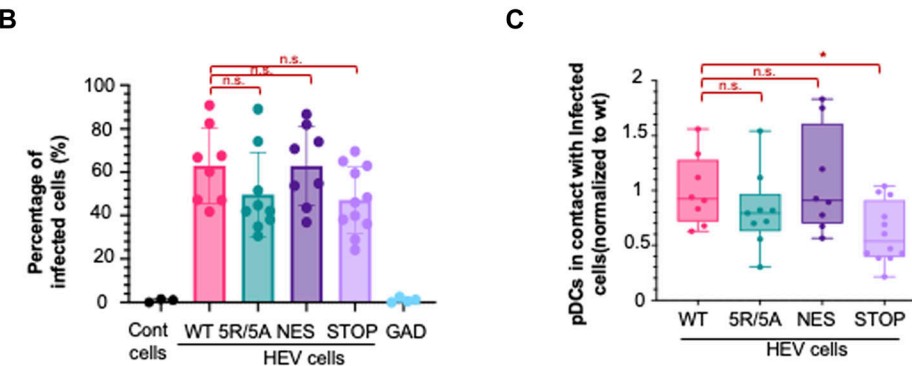

pDCs (Fig 2A; left panel). Single-cell imaging flow cytometry allowed us to distinguish infected cells with (trORF2+) or without ORF2 nuclear translocation (trORF2–). We then examined these cells for IRF3 nuclear translocation, which is known to up-regulate the expression of several cytokines, including IFN-I, IFN-III, and CXCL10 (Brownell et al, 2014), as well as surface molecules. Secretion of chemokines can lead to the recruitment of certain immune cell subsets. For instance, CXCL10 secretion triggers pDC recruitment (Di Domizio et al, 2020).

We observed that around 10% of ORF2+ cells displayed IRF3 nuclear translocation (TrIRF3+) (Fig 6B). Further, around 80% of TrIRF3+ cells exhibited TrORF2+ or an ORF2 nuclear signal (Fig 6C). These results revealed that the immune status of IRF3 in HepG2/C3A-infected cells can differ depending on the nuclear translocation of HEV ORF2, further validating the impact of ORF2 nuclear localization on the immune status of the cells.

Together, our results suggest that ORF2g/c production likely modulates the number of cell–cell contacts and, subsequently,

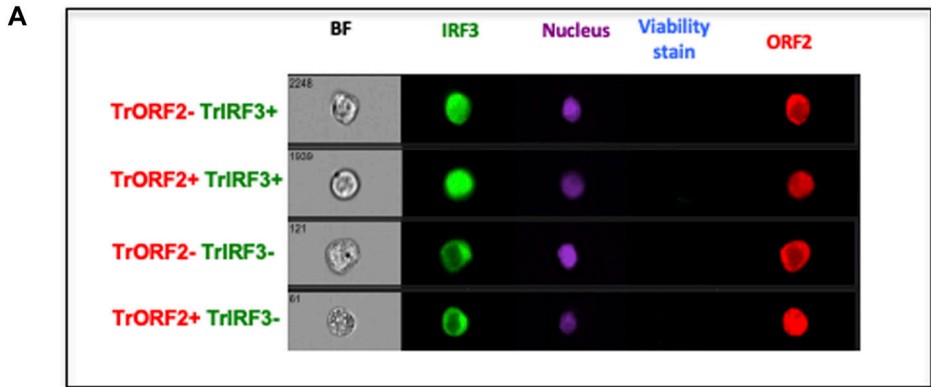

**A**

| | BF | IRF3 | Nucleus | Viability stain | ORF2 |
|---|---|---|---|---|---|
| **TrORF2- TrIRF3+** | | | | | |
| **TrORF2+ TrIRF3+** | | | | | |
| **TrORF2- TrIRF3-** | | | | | |
| **TrORF2+ TrIRF3-** | | | | | |

**Figure 6. Single-cell analysis of IRF3 nuclear localization in HEV-infected HepG2/C3A cells.** HepG2/C3A cells expressing IRF3-GFP were electroporated with WT HEV RNA and fixed 2 d.p.e. after staining with viability stain (Zombie aqua), and were then stained for ORF2 (APC) and nucleus (Hoechst). Single-cell images of viable ORF2+ cells were obtained and categorized according to the localization of IRF3 and ORF2. **(A)** Cells with translocation of ORF2 and IRF3 into the nucleus are represented by TrORF2+ and TrIRF3+. Cells without translocation of ORF2 and IRF3 into the nucleus are represented by TrORF2– and TrIRF3–. **(B)** Percentage of HEV-replicating or ORF2+ cells with nuclear IRF3 translocation (ORF2+ TrIRF3+). **(C)** Percentage of ORF2 nuclear translocation (TrORF2+) among TrIRF3+ cells. Bars represent means ± SD, and each dot represents one independent experiment (n = 4). Statistical analysis was performed using a two-sided Wilcoxon rank-sum test with continuity correction; *P*-values: * ≤ 0.05, ** ≤ 0.005, and *** ≤ 0.0005.

**B** **C**

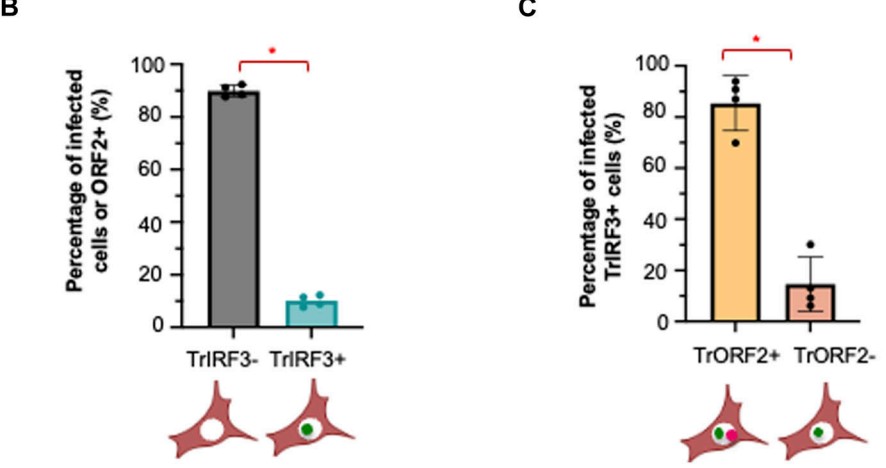

pDC-mediated IFN-I response. In the absence of ORF2g/c (STOP mutant), there is a significant decline in pDC-infected cell contacts. On the other hand, nuclear ORF2 can dictate the immune status in the infected cell via the IRF3 signaling axis, possibly fueling the pDC response.

## Discussion

Several studies have highlighted the suppressive impact of HEV on IFN response (Devhare et al, 2021). Here, we investigated whether these inhibitory mechanisms can be overcome by pDCs, which act as immune sentinels against viral infections. Given the limited number of studies on cellular systems that effectively replicate HEV, we first determined the ability of three hepatoma cell lines to respond to several immune stimuli and replicate the viral genome. Based on their robust response to stimuli and their ability to replicate gt3 p6 strain HEV RNA, we selected PLC3 and HepG2/C3A cells for further analyses. Upon 6 d of HEV RNA electroporation, we observed a modest up-regulation of IRF3- and NF-KB–mediated gene expression in HepG2/C3A cells. The genes that were significantly up-regulated in HEV-replicating HepG2/C3A cells included *ISG15*, *IFN-λ1*, *OAS-2*, and *IL-6*. The up-regulation of *IFN-λ1* expression in HepG2/C3A cells was in agreement with studies

recognizing type III IFN response as a fundamental anti-HEV cytokine in hepatocytes both clinically (Murata et al, 2020) and in vitro (Yin et al, 2017; Wu et al, 2018). Even though HEV can persist in the presence of a sustained type III IFN response (Yin et al, 2017), this response prevents inflammation and helps maintain barrier integrity (Broggi et al, 2020). Importantly, *TNF* up-regulation was reduced by HEV in PLC3 cells, in agreement with previous data obtained in PLC3 cells (Hervouet et al, 2022). This was not observed for HepG2/C3A cells, suggesting that HEV modulates NF-KB–mediated signaling in a cell type–specific manner. While exhibiting strong antiviral effects (Wang et al, 2016), TNF-α is also implicated in proinflammatory processes exacerbating the severity of liver disease upon HEV infection (Behrendt et al, 2017). Importantly, the differences in immune profiles of PLC3 cells and HepG2/C3A may be due to the expression of hepatitis B surface antigen (HBsAg) by PLC3 cells, a known limitation of this cell type (MacNab et al, 1976). Nevertheless, these cells may relevantly mimic the immune characteristics of individuals with preexisting or previous infections or medical condition, predisposing them to HEV infection. In accordance—in the absence of pDC—secreted type III IFNs were undetectable in HEV-replicating PLC3 cells alone, and HEV-replicating cells also did not secrete any detectable IFN-α, consistent with previous studies using HepG2 cells (Yin et al, 2017) and iPSC-HLCs (Wu et al, 2018).

Upon co-culture of HEV-replicating HepG2/C3A cells with pDCs, the three ISGs *MXA*, *ISG15*, and *OAS2*, as well as *IFN-λ1*, were robustly up-regulated. In fact, IFN-III response was also evident at the protein level after 48 h of pDC co-culture with HEV-replicating HepG2/C3A cells. However, the expression of ISGs barely increased at 18 h post-co-culture with HEV-replicating PLC3 cells. Upon pDC co-culture of HEV-replicating PLC3 cells, only *IFN-λ1* was up-regulated significantly at the mRNA level and yet poorly detected as a secreted protein, likely owing to the sensitivity limit of the currently available ELISA detection and thus requiring analysis with more sensitive techniques. *IFN-λ1* expression by pDCs has been documented to actively participate in pDC response, for example, against human cytomegalovirus and SARS-CoV-2 (Yun et al, 2021; Venet et al, 2023). Whether the source of *IFN-λ1* up-regulation was HEV-replicating cells, pDCs, or both cell types in concert remains to be determined. Because the expression of these effectors is crucial for restricting viral replication (Schneider et al, 2014), the absence of ISG up-regulation presents a possible Achilles' heel in host immune defense against HEV as this might contribute to HEV persistence in culture and its elimination might be challenging even after prolonged exposure to high doses of IFNs (Todt et al, 2016). The lack of ISG expression was overcome by pDCs against HEV in the form of a strong IFN-I response when co-cultured with infected cells. This could activate ISG expression in neighboring cells (Bourdon et al, 2020) and also facilitate adaptive immune responses (Crouse et al, 2015). Along the same line, the IFN-I response against HEV was dependent on pDC-infected cell contact formation. This was orchestrated by ICAM-I and $α_Lβ_2$-integrin and TLR7, in agreement with results obtained for other viral infections (Yin et al, 2017). The contact between pDCs and infected cells is beneficial for host defense, considering that HEV is resistant to exogenous IFN treatment (Todt et al, 2016; Wang et al, 2016). On a different note, a prior study has shown that HEV-ORF3 can attenuate TLR7 expression (Lei et al, 2018); this effect is less likely to be observed for pDCs as these are nonpermissive to most viruses (Silvin et al, 2017; Reizis, 2019; Venet et al, 2023). PDC response failed to control HEV replication after 18 h of co-culture with PLC3 cells but effectively controlled HEV replication in HepG2/C3A under the same conditions. We observed more efficient or faster viral control in HepG2/C3A cells compared with PLC3 cells. This was likely due to a significant up-regulation of ISGs (*ISG15*, *IFN-λ1*, and *OAS2*)—even in the absence of pDCs—in HEV-replicating HepG2/C3A cells versus an inexistent one in HEV-replicating PLC3 cells. We hypothesize that the preexisting ISG response in HEV-replicating HepG2/C3A, produced in the absence of pDCs, potentiates a more effective viral control in the presence of pDCs. Therefore, the initial ISG expression in infected cells may synergize the antiviral response by pDCs. Upon further scrutiny, pDC-mediated IFN response reduced HEV production by 50% 48 h post-co-culture in PLC3 cells. This also contributed to an overall decline in viral spread to neighboring cells. These results confirmed the need for a prolonged IFN response for controlling HEV replication.

In the absence of ORF2 nuclear translocation (5R/5A mutant), pDC IFN-I response was diminished in HepG2/C3A cells and abolished in PLC3 cells. Furthermore, when ORF2 accumulated in the nucleus (NES mutant), there was no difference in pDC response as compared to the WT. In addition, the depletion of ORF2g/c secretion (STOP mutant) lowered the pDC response in PLC3 cells but not in HepG2/C3A cells. Altogether, these observations suggest that ORF2 nuclear translocation and secretion of glycosylated ORF2g/c forms might modulate pDC response in a cell type–dependent manner. Because many adhesion molecules are ISGs (Parr & Parr, 2000; Ma et al, 2022a) and can impact the ability of HEV mutant harboring cells to form cell-to-cell contact, we compared their pDC contact–forming propensity. Only cells harboring the STOP mutant differed from WT HEV cells in their ability to form contacts with pDCs. These results support the hypothesis of a reduced attraction of pDCs toward STOP-expressing PLC3 cells, as compared to control cells, because of the absence of ORF2g/c secretion. We did not see the same effect for HepG2/C3A cells, probably because of robust immune induction upon HEV infection. Alternative effector(s) favoring the contacts between pDCs and infected cells could be better expressed in HepG2/C3A cells compared with PLC3 cells, possibly because of the potent immune status, which may mask the effect of ORF2g/c on contact formation. However, this aspect requires further investigation. We should also take into account the decrease in viral infectivity in the absence of ORF2g/c (Fig S2E), which could in turn lead to reduced pDC IFN response. Even though incubation with purified ORF2g/c does not impact HEV entry (Yin et al, 2018), our results do raise further questions on whether its secretion can indirectly offer an advantage to HEV for spreading to the neighboring cell by improving cell–cell adhesion, a relatively easier route of viral transmission (Zhong et al, 2013). Many studies have demonstrated that glycosylation is a common factor affecting cell–cell adhesion (Ohtsubo & Marth, 2006; Gu et al, 2012). When rhesus macaques were infected with the HEV variant that did not express glycosylated ORF2 forms, viral replication was attenuated and viral shedding in feces declined significantly as compared to infection with WT HEV (Ralfs et al, 2023). Therefore, ORF2 g/c may not be essential for viral replication, but it may be instrumental for efficient viral production.

Our findings emphasize the significance of the ongoing co-evolution between hosts and pathogens, allowing the virus to disseminate while concurrently enhancing its detection by immune cells. The difference in pDC response among HEV WT, NES versus 5R/5A still remains enigmatic, but it paves a direction for further exploration in this exciting field. The presence of pDC response to 5R/5A-expressing HepG2/C3A cells may imply regulation of pDC response at an additional level, that is, cell type–dependent regulation. Because pDCs mounted an IFN-I response against 5R/5A in HepG2/C3A cells, we propose that this detection could have been fueled by a successful up-regulation of ISGs within HepG2/C3A cells compared with PLC3 cells upon HEV infection. Importantly, TNF signaling is induced in HepG2/C3A cells and inhibited in PLC3 cells. Because TNF signaling regulates cell adhesion via ICAM-1 (Reglero-Real et al, 2014) and activates an autocrine loop with low and sustained production of type I IFNs (Yarilina et al, 2008), it could contribute to differential pDC response for the two cell types. The difference in pDC-mediated IFN-I response to WT and 5R/5A mutant is particularly interesting and was observed for both cell lines, even though at different magnitudes. The 5R/5A mutant, which is characterized by an absence of nuclear ORF2, displayed a reduced

or abolished pDC response, that is, depending on the cell line. We then uncovered a cell line–specific induction of *TNF*, depending on ORF2 nuclear translocation. It indicates a possible differential modulation of downstream effector(s), including those involved in pDC adhesion/recruitment, and eventually leading to a weaker interferogenic synapse between PLC3 cells and pDCs, but not at play in HepG2/C3A cells. Further investigation will be needed to identify the effector(s) at play. We also investigated whether ORF2 nuclear localization could be involved in the immune state of the infected cells. We showed that IRF3 nuclear translocation tends to occur more in HEV-replicating cells with nuclear ORF2 than in HEV-replicating cells without nuclear ORF2. There could be several possible reasons for this activation of immune signaling, that is, intrusion of nuclear membrane, sensing of the viral ORF2 in the nucleus, or sensing of accidentally shuttled HEV RNA in the nucleus. It would be interesting to investigate this further. The activation of IRF3 signaling could, in turn, aid pDCs in viral sensing via cytokines and up-regulation of other effector molecules.

Globally, the HEV mutants advanced our comprehension about viral regulation of the IFN response and also mechanistic nuances of viral recognition by pDCs. In summary, our findings suggest that pDC response is likely regulated by the (i) transfer of viral entities to pDCs for recognition, (ii) number of cell-to-cell contacts, and (iii) intrinsic immune state of the infected cells.

# Materials and Methods

### Reagents and antibodies

Reagents used for pDC isolation are as follows: Ficoll-Hypaque (GE Healthcare Life Sciences), BDCA-4 magnetic beads (MACS Miltenyi Biotec), LS Columns (MACS Miltenyi Biotec), and bottle-top vacuum filters 0.22 $\mu$m (Nalgene). Other reagents included polyI:C (LMW; InvivoGen) as TLR3 agonist, imiquimod as TLR7 agonist (InvivoGen), and TLR7 antagonist (IRS661, 5'-TGCTTGCAAGCTTGCAAGCA-3') synthesized on a phosphorothioate backbone (MWG Biotech). The following antibodies were used: mouse anti-$\alpha_L\beta_2$ integrin (clone 38; Antibodies Online), mouse anti-ICAM-1 (clone LB-2; BD Bioscience), mouse anti-HEV ORF2 1E6 (antibody registry #AB-827236; Millipore), mouse anti-ORF2i/g/c P3H2 (Bentaleb et al, 2022), and mouse anti-ORF2i P1H1 (Bentaleb et al, 2022). Goat anti-Mouse IgG2b Cross-Adsorbed Secondary Antibody Alexa Fluor 488 (Catalog #A-21141; Thermo Fisher Scientific) was used for flow cytometry analysis.

### Cell lines

PLC3 cells, a subclone of PLC3/PRF/5 cells (Montpellier et al, 2018), Huh-7.5 cells (RRID:CVCL_7927), which are hepatoma cells derived from Huh-7 cells (Blight et al, 2002), and HepG2/C3A cells (ATCC CRL-3581) (provided by Dr. V.L. Dao Thi and Dr. D. Moradpour) were used. PLC3 and Huh-7.5 cells were maintained in DMEM supplemented with 10% FBS, 100 U/ml penicillin, 100 mg/ml streptomycin, 2 mM L-glutamine, nonessential amino acids, and 1 mM sodium pyruvate (Life Technologies) at 37°C/5% $CO_2$. HepG2/C3A cells were maintained in DMEM (GlutaMAX, Pyr-) supplemented with 10% FBS, 100 /ml penicillin at 37°C/5% $CO_2$.

### pDC isolation and culture

pDCs were isolated from blood from healthy adult human volunteers, which were obtained from the "Etablissement Francais du Sang" (EFS; Auvergne-Rhone-Alpes, France) under the convention EFS 16–2066 and according to procedures approved by the EFS committee. Informed consent was obtained from all subjects in accordance with the Declaration of Helsinki. Information on sex and age was available for all subjects, yet we previously showed that pDC responses are within the same range for all donors Décembre et al, 2014. PBMCs were isolated using Ficoll-Hypaque density centrifugation, and pDCs were positively selected from PBMCs using BDCA-4-magnetic beads (MACS Miltenyi Biotec), as previously described (Dreux et al, 2012; Decembre et al, 2014; Venet et al, 2023). The typical yields of PBMCs and pDCs were 500–800 × $10^6$ and 1–2 × $10^6$ cells, respectively, with a typical purity of >95% pDCs. Isolated pDCs were maintained in RPMI 1640 medium (Life Technologies) supplemented with 10% FBS, 10 mM Hepes, 100 U/ml penicillin, 100 mg/ml streptomycin, 2 mM L-glutamine, nonessential amino acids, and 1 mM sodium pyruvate at 37°C/5% $CO_2$.

### Stimulation of cells for immune response using agonists and recombinant cytokines

PLC3 cells were seeded in 24-well plates at a concentration of 8.104 cells/well. After 24 h, cells were incubated with polyI:C LMW in complete medium at a concentration of 25 $\mu$g/ml and 100 $\mu$g/ml, in a total volume of 0.5 ml medium. Alternatively, cells were transfected with polyI:C LMW at a concentration of 0.3 $\mu$g/ml and 2 $\mu$g/ml. Lipofectamine 2000 (Invitrogen) in Opti-MEM I reduced serum media was used for transfection according to the manufacturer's protocol. IFN-$\beta$ subtype 1a (Catalog #PHC4244; Invitrogen), IFN-$\lambda$3/IL-28B (IL-28B) (PBL IFN Source #11820-1), and TNF (#16769; Cell Signaling Technology) were added at a concentration of 100 U/ml, 100 ng/ml, and 535 U/ml, respectively, in a total volume of 0.5 ml medium per well. The cells were harvested 6 h post-treatment for RNA extraction.

### Analysis of transcriptional levels by RT–qPCR

RNAs were isolated from cells harvested in guanidinium thiocyanate citrate buffer (GTC) by phenol/chloroform extraction procedure, as described previously (Assil et al, 2019a). The mRNA levels of human *MXA*, *ISG15*, *IFN-λ1*, *IL-6*, *TNF*, *ORF2*, and glyceraldehyde-3-phosphate dehydrogenase (*GADPH*) were determined by RT–qPCR using High-Capacity cDNA Reverse Transcription Kit (ref# 4368813) and PowerUp SYBR Green Master Mix (ref# A25778) for RT–qPCR run on QuantStudio 6 PCR system and analyzed using QuantStudio Design and Analysis Software (Thermo Fisher Scientific). The mRNA levels were normalized to *GADPH* mRNA levels. The sequences of the primers used for RT–qPCR are listed in Table S1.

 **Life Science Alliance**

## Plasmids and electroporation for HEV infectious system

The plasmid pBlueScript SK(+) carrying the DNA of the full-length genome of adapted gt3 Kernow C-1 p6 strain (GenBank accession number JQ679013, kindly provided by Dr SU Emerson) was used as the WT genome. The 5R/5A and NES mutant plasmids were previously described (Hervouet et al, 2022). The STOP mutant was generated by site-directed mutagenesis using the following primers:

 P6/mutstop-F: TGTTCTGCTGCTGTagTTCGTGTTTCTGCCTATGC.
 P6/mutstop-R: GGCAGAAACACGAACtACAGCAGCAGAACAACCC.

The Stop mutation was introduced by sequential PCR steps, as previously described (Ankavay et al, 2019), and verified by DNA sequencing. The non-replicative GAD mutant plasmid was a gift from Dr VL Dao Thi (Emerson et al, 2013). To prepare genomic HEV RNAs (capped RNA), the WT and mutant pBlueScript SK(+) HEV plasmids were linearized at their 3′ end with the MluI restriction enzyme (NEB) and transcribed with the mMESSAGE mMACHINE T7 Transcription kit (#1344; Ambion). 10 $\mu$g capped RNAs were next delivered to cells by electroporation using Gene Pulser Xcell 617BR 11218 (Bio-Rad).

## Co-culture of infected cells with isolated pDCs and ELISA

Unless otherwise indicated, $5 \times 10^4$ pDCs were co-cultured with $6 \times 10^4$ PLC3 or HepG2/C3A cells, electroporated or not, or, as a control condition, filtered supernatants (0.4 $\mu$m). Infected cells were electroporated for 48, 72, or 144 h (as indicated) in a 200 $\mu$l final volume in 96-well round-bottom plates incubated at 37°C/5% $CO_2$. When indicated, cells were co-cultured in 96-well format transwell chambers (Corning) with a 0.4-$\mu$m permeable membrane. 16–18 h later (as specified), cell culture supernatants were collected and the levels of IFN-$\alpha$ were measured using a commercially available specific ELISA kit (PBL IFN Source Catalog #411052) with an assay range of 12.5–500 pg/ml.

## Flow cytometry–based viral spread assays and imaging flow cytometry by ImageStreamX technology

PLC3 cells were transduced with lentiviral-based vector pseudotyped with vesicular stomatitis virus glycoprotein to stably express GFP. After immunoisolation, pDCs were stained for 20 min at 37°C in the dark. Labeled pDCs were then spun down and resuspended in pDC culture medium. $5 \times 10^4$ pDCs were co-cultured with $3 \times 10^4$ HEV-replicating cells (electroporated 6 d before co-culture) and with $3 \times 10^4$ GFP+-uninfected cells for 48 h at 37°C/5% $CO_2$. The level of viral spread from HEV-replicating cells (ORF2+) to uninfected cells (GFP+) during co-culture was determined by flow cytometry analysis as the frequency of infected cells (ORF2+/GFP+ population) among the GFP+ cell population and similarly in GFP- populations. At the indicated times, harvested cells were resuspended using 0.48 mM EDTA/PBS solution for the co-culture with pDCs. Cells were then incubated with 1 $\mu$l/ml viability marker diluted in PBS for 20 min at RT. Cytoperm/ Cytofix and permeabilization/wash solutions (BD Bioscience) were used in the subsequent stages of the ORF2 staining protocol optimized in-house for flow cytometry. Cells were fixed with Cytofix for 20 min at 4°C and were then washed twice and resuspended in 1x Permwash. The fixed cells were treated with cold methanol (−20°C)

for 45 min and washed twice with 1x Permwash. The cells were then incubated with 1E6 anti-HEV antibody for a minimum of 3 h. After another wash, cells were incubated with a secondary antibody, that is, Goat anti-Mouse IgG2b Cross-Adsorbed Secondary Antibody Alexa Fluor 488 (Catalog #A-21141; Thermo Fisher Scientific) for 2 h. The samples were acquired after final two washes. Flow cytometry analysis was performed using Canto II Becton Dickinson using BD FACSDIVA v8.1 software, and the data were analyzed using Flow Jo 10.8.1 software (Tree Star). The CTV Cell Proliferation kit (Life Technologies) was used to stain pDCs. LIVE/DEAD Fixable Near-IR Dead Cell Stain Kit (ref #L10119; Thermo Fisher Scientific) was used to check viability. For imaging flow cytometry, cells expressing IRF3-GFP (construct was kindly provided by Dr. Marco Binder) (Willemsen et al, 2017) were electroporated with HEV WT RNA. The cells were then harvested at 48 h.p.e., followed by viability staining with Zombie Aqua (Cat # 423101; BioLegend), and then ORF2 staining protocol. At the final step, Hoechst (ref #H1399; Thermo Fisher Scientific) was used to stain the nuclei of the cells. After acquisition by ImageStreamX Mark II (Amnis—Millipore), analysis was done using IDEAS software.

## Microscopy and analysis of images

At 4 h post-co-culture, CTV-stained pDCs and infected PLC3 and HepG2/C3A cells were fixed with PFA 4%, followed by immunostaining using anti-ORF2 P3H2 antibodies with a protocol optimized previously (Bentaleb et al, 2022). Confocal imaging was performed using Zeiss LSM980 scanning confocal microscopy. Co-cultured cells were automatically segmented based on the CMTPX staining for the cell line used and CTV labeling for pDCs. Next, the infected cells were identified by ORF2 expression, and number/frequency of infected cells were quantified using one of two similar versions of a home-made ImageJ macro (https://github.com/jbrocardplatim/PDC-contacts), adapted to HepG2/C3A or PLC3 cells. The frequency of pDCs within 3 $\mu$m of any cell was calculated, as well as the number of pDCs within 3 $\mu$m of any infected cell (from 0 to 3+). An additional macro was also used to measure the frequency of pDCs closer than 1 $\mu$m to infected cells and/or in direct contact; the individual contact areas (in $\mu$m$^2$) were measured as well (https://github.com/jbrocardplatim/PDC-contacts). The calculation formula for contact frequencies considered variations in the number of infected cells to mitigate potential biases:

$$\frac{\text{Number of pDC – infected cell contact}}{\text{Isolated pDCs + isolated infected cells + pDC – infected cell contact}}.$$

To compare the ORF2 localization in mutants and WT bearing cells, nuclei were stained with NucSpot Live 650 (Catalog #40082; Biotium) and HEV-ORF2 was stained with 1E6 antibody. For all microscopy experiments, cells were seeded in 96-Well Optical-Bottom Plate (Thermo Fisher Scientific), coated with poly-L-lysine (P6282; Sigma-Aldrich).

## Western blotting and immunoprecipitation analyses

Western blotting (WB) and immunoprecipitation (IP) analyses were performed as described previously (Bentaleb et al, 2022). For WB,

supernatants and lysates of WT- and STOP-expressing PLC3 cells were heated for 20 min at 80°C in the presence of reducing Laemmli buffer. Samples were then separated by 10% SDS–PAGE and transferred onto nitrocellulose membranes (Hybond ECL). ORF2 proteins were detected with 1E6 antibody and peroxidase-conjugated anti-mouse antibodies. The detection of proteins was done by chemiluminescence analysis (ECL). For IP, P3H2 and P1H1 antibodies were bound to M-280 Dynabeads (Thermo Fisher Scientific) overnight at 37°C following the manufacturer's recommendations. Beads were washed and then incubated for 1 h at room temperature with supernatants (heat-inactivated) or cell lysates. Beads were washed and then heated at 80°C for 20 min in reducing Laemmli buffer. ORF2 proteins were detected by WB using the 1E6 antibody.

## Statistical analysis

Statistical analysis was performed using R software environment for statistical computing and graphics (version 3.3.2). For quantifications by ELISA, RT–qPCR, and flow cytometry, analyses were performed using the Wilcoxon rank-sum exact test and *P*-value adjustment method: Bonferroni or FDR was applied when mentioned. The figures were prepared using PRISM software (version 10.2.1).

# Supplementary Information

# Acknowledgements

We would like to thank ANRS-MIE for providing financial support (ANRS-MIE grant number: AAP2020-2/ECTZ133955) and "Contrats doctoraux Lyon 1 dédiés à l'International" from Université Lyon 1 for G Joshi's PhD fellowship. This work was also supported by grants from the Agence Nationale de la Recherche (ANRJCJC-iSYN); the Agence Nationale pour la Recherche contre le SIDA et les Hépatites Virales (ANRS—N21006CR and N19017CR); and the UDL/ANR IA ELAN ERC (G19005CC) to M Dreux. We acknowledge Dr. Julie Lucifora, Dr. David Durantel, Dr. Elena Tomasello, Dr. Alexandre Belot, Dr. Søren Riis Paludan, Dr. Marco Binder, and Dr. Antoine Marçais for scientific discussions. We thank SFR Biosciences (UMS3444/CNRS, US8/Inserm, ENS de Lyon, UCBL) including the PLATIM and AniRA-cytometry facilities, for technical assistance in imaging and FACS analyses, respectively. Special thanks to Dr. Marion Delphin, Dr. Vladimir Goncalves Magalhaes, and Roxanne Fouille for their contribution in supporting experiments. The contribution of the EFS Confluence/Decine-Lyon is also noteworthy. Finally, we thank current and former members of the VIV team for helpful discussions and support: Dr. Margarida Sa Ribeiro, Célia Nuovo, Matteo Agostini, and Manon Venet.

## Author Contributions

G Joshi: conceptualization, data curation, formal analysis, funding acquisition, investigation, methodology, and writing—original draft, review, and editing.
E Décembre: investigation and methodology.
J Brocard: formal analysis, validation, and methodology.
C Montpellier: resources, investigation, and methodology.
M Ferrié: resources, investigation, and methodology.
O Allatif: formal analysis and validation.
A-K Mehnert: investigation.
J Pons: investigation.
D Galiana: formal analysis, supervision, investigation, and writing—review and editing.
VL Dao Thi: resources, supervision, methodology, and writing—review and editing.
N Jouvenet: formal analysis, supervision, validation, and writing—review and editing.
L Cocquerel: conceptualization, resources, formal analysis, supervision, validation, methodology, and writing—review and editing.
M Dreux: conceptualization, supervision, funding acquisition, and writing—original draft, review, and editing.

## Conflict of Interest Statement

The authors declare that they have no conflict of interest.

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
