## [Reviewer comments · Life Science Alliance]

Life Science Alliance

Plasmacytoid dendritic cell sensing of Hepatitis E Virus is shaped by both viral and host factors

Garima Joshi, Elodie Décembre, Jacques Brocard, Claire Montpellier, Martin Ferrié, Omran Allatif, Ann-Kathrin Mehnert, Johann Pons, Delphine Galiana, Viet Loan Dao Thi, Nolwenn Jouvenet, Laurence Cocquerel, and Marlene Dreux

DOI: <https://doi.org/10.26508/lsa.202503256>

Corresponding author(s): Garima Joshi, Centre International de Recherche en Infectiologie

Review Timeline:	Submission Date:	2025-02-06
	Editorial Decision:	2025-02-24
	Revision Received:	2025-03-14
	Accepted:	2025-03-14

Transaction Report:

Please note that the manuscript was reviewed at *Review Commons* and these reports were taken into account in the decision-making process at *Life Science Alliance*.

Review
COMMONS

Review #1

Plasmacytoid dendritic cells (pDCs) are the major producers of type I interferon after viral infections and play key role in antiviral immune response. This article by Joshi et al. investigates the role of pDCs in regulating the Hepatitis E virus (HEV) infection. In Fig. 1, the authors investigated the immunocompetence of different cell lines and HepG2/C3A and PLC3 were chosen for further studies. By utilizing a combination of flow cytometry, RT-qPCR and other techniques, the authors showed in Fig. 2 that the cell-cell contacts between pDCs and HEV infected cells induce the pDCs to secrete interferon (IFN). This interaction is mediated by cell adhesion molecules and is dependent on TLR7 signaling. The authors then went on to show that the IFN produced by pDCs controlled the viral spread. Further, using several mutant forms of ORF2 protein and utilizing imaging, RT-qPCR and other techniques, in Fig. 3 and 4 the authors elucidated the importance of the glycosylation pattern, localization of different forms of HEV ORF2 protein, cell-cell contact in triggering the immune response. Overall, this study provided insights in the pDC mediated IFN response against HEV.

Major comments:

1. The authors report that in the PLC3 cells, STOP mutation significantly reduced IFN α production (Fig. 3f), significantly reduced pDC contact with infected cells (Fig. 4c) and thus concluded that the ORF2g/c is involved in pDC-infected cell interaction and IFN α production. However, in the HepG2/C3A cells, the STOP mutation does not decrease the IFN α production (Fig. 3e). In the manuscript, one of the key conclusions is that the glycosylated form of ORF2 leads to better recognition of the infected cells by pDC. So, it is critical that the difference in the IFN α production between these two cell lines with STOP mutation is addressed with further details.
2. The authors show that the IFN α response was reduced in 5R/5A mutant HepG2/C3A cells (Fig. 3e), whereas the IFN α response was completely absent in 5R/5A mutant PLC3 cells (Fig. 3f). The authors suggested that the difference in IFN α response may be due to lack of ORF2i in PLC3 and other cell specific regulation in HepG2/C3A. Further evidence for this differential regulation would strengthen the claim.
3. In the PLC3-pDC co-culture experiment (Fig. 2b), there is already an induction of IFN- λ 1 (Interferon Lambda 1) in the uninfected PLC3-pDC co-culture (right panel, Fig. 2b). An explanation for the IFN- λ 1 (Interferon Lambda 1) expression in the uninfected state would be helpful.

Additional comments:

1. Authors checked the expression of two ISGs- MXA, ISG15 in Fig. 1a-c, 2a-b. Were the expressions of other ISGs, such as members of OAS family (OAS1, OAS2 etc.), IFITM family or any other ISGs checked? This may be helpful, since in the Fig. 2c there is IFN α production in pDC-infected PLC3 co-culture, but the ISGs (MXA, ISG15) are not upregulated significantly in Fig. 2b.
2. In the HepG2/C3A-pDC co-culture experiment (Fig. 2a), there is not much difference in IFN- λ 1 (Interferon Lambda 1) level in the infected HepG2/C3A-pDC co-culture (right panel, Fig. 2a) in comparison to infected HepG2/C3A alone (left panel, Fig. 2b), and also this outcome is different from that in the PLC3 experiment (Fig. 2b). Further clarification would help to support the conclusion regarding the IFN- λ 1 (Interferon Lambda 1) upregulation in HEV infected cells-pDC co-culture.
3. The authors show that in the pDC-PLC3 co-culture system, IFN α was induced at 18h (Fig. 2c-2e), but the viral replication was not decreased in PLC3 cells (Fig. 2g). But, the HepG2/C3A-pDC co-culture has reduced viral replication at 18h (Fig. 2f). An explanation for the difference in the observation in two different cell lines at the same timepoints would strengthen the antiviral role of pDCs on HEV infected cells.
4. The authors quantified the fold change in HEV infected PLC3+ cells in Fig. 2h. Was it performed by flow cytometry? It would be helpful to mention it in the figure legend. Also, if the said quantitation was done by flow cytometry, performing similar assay with HEPG2/C3A cells at 48h would provide the readers a better idea about the antiviral response across the cell lines at comparable timepoints.

****Minor comments:****

1. Was it expected to observe the increased induction of IL6 (Fig. 1b) in HepG2/C3A cells (but not in other cell lines) after IFN- λ (Interferon Lambda) treatment?
2. In Fig. 3e, for the WT cells, 4 datapoints are visible while in the legend it is mentioned n=5.
3. Typo: IRS661 in line 263, 699, Figure 2e.
4. Typo: 200 μ l in line 579.
5. Catalogue number for ELISA kit is missing (Line 584).
6. It would be helpful if the color code for the imaging in Supplementary figure 2f is provided on the top of the images, as it is provided in other images.

This article by Joshi et al. provides insight about the role of pDCs in controlling the HEV infection. However, the importance of pDC-infected cell contact mediated IFN-I secretion in antiviral response has been previously shown by the authors' group (Assil et al., 2019, Cell Host & Microbe) and others as well (E.g., Yun et al., 2021, Sci. Immunol.). The involvement of integrin mediated cell adhesion and TLR signaling in mediating this response was also shown. Though this manuscript does not advance the field of pDC biology or virology significantly, it does provide better understanding of the pDC antiviral response in the landscape of HEV infection. Although, it is out of the scope of this manuscript, elucidation of the mechanistic regulation how ORF2g/c controls the pDC-infected cell contact would be of great interest and significance. Overall, this study could be of interest to a general audience, especially to the virologists and researchers working in pDC biology.

Review #2

Joshi et al. discovered that human pDC isolated from blood of healthy donors incubated in coculture with hepatitis E virus (HEV) infected hepatoma cell lines were triggered in a cell-cell contact-dependent manner to mount interferon-alpha responses. This conclusion was supported by experiments in a Transwell setting in which pDC were separated from infected cells by a permeable membrane and under which conditions pDC were not triggered to mount interferon-alpha responses. Treatment of the coculture system with an TLR7 inhibitor or with antibodies against the cell adhesion receptors ICAM-I and alphaLbeta2-integrin either entirely inhibited or reduced the induction of interferon-alpha responses, respectively. HepG2/C3A cells infected with different ORF2 mutants induced overall similar interferon-alpha responses in pDC. In contrast, PLC3 cells infected with an ORF2 variant that is not properly translocated into the nucleus did not induce interferon-alpha responses in pDC, whereas the other variants induced normal responses. This observation indicated that in the context of PLC3 cells the subcellular localization of ORF2 in the nucleus is critical to induce interferon-alpha responses in pDC. Finally, quantification of contact points between pDC and infected cells supported the conclusion that enhanced cell-cell contact was necessary to efficiently induce interferon-alpha responses in pDC.

Overall, the study was carefully carried out and shows interesting results. It is appreciated that responses of human pDC isolated from blood of healthy donors were analyzed. Furthermore, the stimulation of pDC with infected hepatoma cell lines is interesting. Earlier studies showed that infected cells might be better pDC stimulators than free virus. Furthermore, in line 140 of the manuscript the authors correctly state that pDCs are resistant to virtually all viruses. Thus, they should have included experiments in which they stimulate pDC with free HEV. Comparative analysis of the data presumably would further highlight the relevance of pDC triggering by infected cells.

Despite interesting observations are presented regarding differences in the induction of pDC responses with PLC3 and HepG2/C3A cells infected with an ORF2 variant that is not properly translocated into the nucleus, no experiments were offered to explain this phenomenon. It is highly recommended to include additional experiments that support concepts either in one or the other direction as presented in the discussion.

The paper is written in an unnecessarily complicated way. The authors should try to arrange the manuscript in a manner that the readability is enhanced. In the end, the observations that have to be communicated are not very complicated. Showing only controls in Fig. 1 is a cumbersome start in the manuscript. The authors should

consider moving such results in a supplementary Figure. The authors do not thoroughly describe the experiment in Fig. 2I in the results section, although it seems to be rather interesting. The authors should give more explanations in the results section how they quantified cell-to-cell HEV infection in presence or absence of pDC.

****Minor comments****

In the legend of Fig. 1D and E it should be clarified what "Cont cells" and "HEV cells" means. I assume it is the mock control and HEV infected cells. It should be mentioned in the figure caption with which multiplicity of infections the cells were treated.

For reasons of consistency, in Fig. 2 controls should be identified also by "Cont cells" and not by "cont cell" (in A and B) and "cont cells" (in C and D).

In Fig. 2H, above "Fold-change in ORF2+ cells" there is indicated PLC3 in small letters. Should this be better moved above the panel, as done in F and G?

The schematic depiction of different cell types in Fig. 2H and J is not very helpful, if not explained in the figure caption.

In Fig. 3B the labels shown on the left side of A should be repeated.

Why above the second row in A and B it is indicated "ORF2-AF488", but everywhere else in the manuscript it is referred to ORF2. If the AF488 addition is necessary (presumably it refers to an Alexa Fluor 488 conjugate of the antibody used to detect ORF2), this should be introduced either in the Materials and Methods section or in the figure caption.

The study is of high relevance. The presented phenomena are clearly described. Mechanistic aspects are not fully covered, yet. This study will be interesting for a relatively broad audience of virologists and immunologists. Currently, the manuscript is written in a manner that it is relatively difficult to read. This should be improved to reach the relatively divergent audience.

Corresponding author(s): Garima Joshi, Marlène Dreux

1. General Statements [optional]

We sincerely thank the reviewers for their constructive and insightful comments, which have greatly enhanced the quality and clarity of our manuscript. Novel experiments have been performed in an attempt to clarify all the points they raised (new figures and new supplemental data have been produced). Each reviewer's comment has been addressed separately and changes are marked in red in the text ('see also "the point-by-point response" below).

This section is mandatory. Please insert a point-by-point reply describing the revisions that were already carried out and included in the transferred manuscript.

Reviewer #1

Evidence, reproducibility and clarity

Plasmacytoid dendritic cells (pDCs) are the major producers of type I interferon after viral infections and play key role in antiviral immune response. This article by Joshi et al. investigates the role of pDCs in regulating the Hepatitis E virus (HEV) infection. In Fig. 1, the authors investigated the immunocompetence of different cell lines and HepG2/C3A and PLC3 were chosen for further studies. By utilizing a combination of flow cytometry, RT-qPCR and other techniques, the authors showed in Fig. 2 that the cell-cell contacts between pDCs and HEV infected cells induce the pDCs to secrete interferon (IFN). This interaction is mediated by cell adhesion molecules and is dependent on TLR7 signaling. The authors then went on to show that the IFN produced by pDCs controlled the viral spread. Further, using several mutant forms of ORF2 protein and utilizing imaging, RT-qPCR and other techniques, in Fig. 3 and 4 the authors elucidated the importance of the glycosylation pattern, localization of different forms of HEV ORF2 protein, cell-cell contact in triggering the immune response. Overall, this study provided insights in the pDC mediated IFN response against HEV.

Major comments

1. The authors report that in the PLC3 cells, STOP mutation significantly reduced IFN α production (Fig. 3f), significantly reduced pDC contact with infected cells (Fig. 4c) and thus concluded that the ORF2g/c is involved in pDC-infected cell interaction and IFN α production. However, in the HepG2/C3A cells, the STOP mutation does not decrease the IFN α production (Fig. 3e). In the manuscript, one of the key conclusions is that the glycosylated form of ORF2 leads to better recognition of the infected cells by pDC. So, it is critical that the difference in the IFN α production between these two cell lines with STOP mutation is addressed with further details.

We agree that our conclusion about the involvement of ORF2g/c in pDC response stands only for PLC3 cells. To ensure that the HEV STOP mutant was deficient in ORF2g/c secretion in both cell lines, we examined its secretion by western blot in cell culture supernatants. These analyses confirmed the absence of ORF2g/c secretion by the STOP mutant from both PLC3 and HepG2/C3A cells (**Supplementary Figure 2**). Therefore, the differences observed between PLC3-STOP and HepG2/C3A-STOP are not due to differential secretion of ORF2g/c by the STOP mutant but may be attributed to differential immune signaling in the two cell types.

Supplementary Figure 2. Production of ORF2g/c forms. (G) Western blot analysis of ORF2 in the supernatants of HEV-replicating HepG2/C3A and PLC3 cells 6d.p.e. in two independent experiments (using 1E6 mAb, which detects all ORF2 species).

We have added the description of this novel data in the results section (Line 338):

It is noteworthy that we observed the absence of ORF2g/c production in cell culture supernatants of the STOP mutant expressed in both PLC3 and HepG2/C3A cells (Supplementary Figure 2g).

PLC3 and HepG2/C3A cells differ for their innate immune responsiveness. For instance, the magnitude of the ISG response was lower in PLC3 cells than in HepG2/C3A cells when infected with HEV (Figure 2a-b, left panels). This difference can contribute to differences in expression of signaling proteins at the infected cell-pDC contacts, leading to a quantifiable effect on PLC3 cells, while not in HepG2/C3A cells. The dissimilarities in pDC response to the mutants among the two cell types can likely be attributed to the differential activation of immune response in HEV-infected cells,

providing a broad view (and not limited to one cell type) of immune regulation of pDC response. Even though the role of glycosylated ORF2g/c forms in immunomodulation cannot be generalized for all cell types, it is important to highlight their effect on PLC3-pDC response due to the pathogenic outcome of HEV in immunosuppressed patients.

We have emphasized this point in the new version of the discussion (lines 530-536):

These results support the hypothesis of a reduced attraction of pDCs towards STOP-expressing PLC3 cells, as compared to control cells, due to the absence of ORF2g/c secretion. We did not see the same effect for HepG2/C3A cells, probably due to robust immune induction upon HEV infection. Alternative effector(s) favoring the contacts between pDCs and infected cells could be better expressed in HepG2/C3A cells compared to PLC3 cells, possibly due to the potent immune status, which may mask the effect of ORF2g/c on contact formation. However, this aspect requires further investigation.

2. The authors show that the IFN α response was reduced in 5R/5A mutant HepG2/C3A cells (Fig. 3e), whereas the IFN α response was completely absent in 5R/5A mutant PLC3 cells (Fig. 3f). The authors suggested that the difference in IFN α response may be due to lack of ORF2i in PLC3 and other cell specific regulation in HepG2/C3A. Further evidence for this differential regulation would strengthen the claim.

Our results showed a reduction of pDC response when the 5R/5A mutant was expressed, as compared to the wild-type protein, both in PLC3 and HepG2/C3A cells. This reduction was more pronounced in PLC3 cells as compared to HepG2/C3 cells. Although this reduction was statistically significant in HepG2/C3A, our results uncovered that the intensity of ORF2i-mediated regulation of pDC response might be modulated by cell type-specific alterations upon HEV infection. Similar to our response to point 1, we proposed that this can result, at least in part, from cell type-dependent differences in inflammatory/innate immunity responsiveness among the two cell types. Notably, we observed that HEV infection differentially impacts the expression of *Tumor necrosis factor (TNF)*, a representative cytokine of *NFKB*-induced signaling (**Figure 2a**). Interestingly, *TNF* is known to impact the expression of regulators of cell adhesion and/or immune cells recruitment, including *ICAM1* (<https://doi.org/10.3858/emm.2008.40.2.167>) and various inflammatory chemokines (<https://doi.org/10.1016/j.cytogfr.2014.07.016>).

We thus decided to analyze how 5R/5A mutation impacts *TNF* expression in the two cell lines (**Supplementary figure 3**). We found that 5R/5A mutant significantly reduced *TNF* induction in PLC3 cells but not in HepG2/C3A cells as compared to HEV WT. Our results thus show a cell type-specific regulation of *TNF* expression – potentially along with *NFKB*-induced signaling - depending on the nuclear translocation of ORF2, implying potential differential modulation of downstream effectors, including those involved in pDC adhesion/recruitment (e.g. *ICAM*), and hereby leading to weaker contacts between PLC3 cells and pDCs *versus* HepG2/C3A cells and pDCs in absence of ORF2 nuclear translocation.

The following is now inserted in the result section (Lines 385-397):

Here, we showed that the pDCs exhibited reduced $IFN\alpha$ response to HepG2/C3A expressing the 5R/5A mutant, and a complete absence of this response in PLC3 cells. This is likely due to differences in immune signaling among the two cell lines with the 5R/5A mutant. We observed that HEV infection impacted, in a cell-type dependent manner, the expression of TNF (Figure 2a), a representative cytokine of NFKB signaling, known to modulate the expression of regulators of pDC adhesion and/or recruitment, including ICAM1 (Kim et al., 2008) and various inflammatory chemokines (Sedger & McDermott, 2014). We thus decided to analyze the effect of 5R/5A mutation on TNF expression in these two cell lines (Supplementary figure 3a-b). We found that 5R/5A mutant significantly reduced TNF induction in PLC3 cells but not in HepG2/C3A cells as compared to HEV-WT, thus potentially explaining the cell type-specific regulation by ORF2i. Taken together, our results show that ORF2 protein expression and localization in producer cells modulate pDC response, with a magnitude that is cell-type dependent.

The following is now inserted in the discussion section (Lines 562-570):

The difference in pDC-mediated $IFN-I$ response to WT and 5R/5A mutant is particularly interesting and was observed for both cell lines, even though at different magnitudes. The 5R/5A mutant, which is characterized by an absence of nuclear ORF2, displayed a reduced or abolished pDC response i.e., depending on the cell line. We then uncovered a cell line-specific induction of TNF, depending on ORF2 nuclear translocation. It indicates a possible differential modulation of downstream effector(s), including those involved in pDC cell adhesion/recruitment, and eventually leading to a weaker interferogenic synapse between PLC3 cells and pDCs, but not at play in HepG2/C3A cells. Further investigation will be needed to identify the effector(s) at play.

Novel supplementary Figure 3. Abundance of *TNF* expression in HepG2/C3A and PLC3 cells upon HEV replication. Quantification of the transcript levels of *TNF* determined at 6 d.p.e. in HepG2/C3A (A) and PLC3 (B) cells. Bars represent copy number per µg total RNA; means ± SD; each dot represents one independent experiment (n=5 to 7). Statistical analysis

was done using Wilcoxon rank sum test with continuity correction; p values as: * ≤ 0.05 , ** ≤ 0.005 and *** ≤ 0.0005 .

To further investigate the effect of ORF2 nuclear localization on immune signaling, we carried out single-cell imaging flow cytometry experiments in HepG2/C3A cells, which demonstrated more efficient upregulation of ISGs than in PLC3 cells, therefore also higher probability of IRF3 nuclear localization. This approach allowed to distinguish infected cells with (trORF2+) or without ORF2 nuclear translocation (trORF2-) (**Figure 6a**). We then examined these cells for IRF3 nuclear translocation, which is known to upregulate the expression of several cytokines, including IFN-I, IFN-III, and CXCL10 (doi: 10.1128/JVI.02007-13), as well as surface molecules. Secretion of chemokines can lead to the recruitment of certain immune cell subsets. For instance, CXCL10 secretion triggers pDC recruitment (<https://doi.org.proxy.insermbiblio.inist.fr/10.1038/s41590-020-0721-6>).

Novel figure 6a. Single-cell analysis of IRF3 nuclear localization in HEV infected HepG2/C3A cells. HepG2/C3A cells expressing IRF3-GFP electroporated with WT HEV RNA and fixed 2 d.p.e. after staining with viability stain (Zombie aqua), ORF2 (APC) and DNA (Hoechst). Single-cell images of viable ORF2+ cells were obtained and categorized according to the localization of IRF3 and ORF2. (**A**) Cells with translocation of ORF2 and IRF3 into the nucleus are represented by TrORF2+ and TrIRF3+. Cells without translocation of ORF2 and IRF3 into the nucleus are represented by TrORF2- and TrIRF3-.

We observed that around 10% of trORF2+ cells displayed IRF3 nuclear translocation (TrIRF3+) (**Figure 6b**). Around 80% of TrIRF3+ cells exhibited an ORF2 nuclear signal (**Figure 6c**). These results revealed that the immune status of IRF3 in HepG2/C3A infected

cells can differ depending on the nuclear translocation of HEV ORF2, further validating the impact of ORF2 nuclear localization on the immune status of the cells.

Figure 6b and c. Single-cell analysis of IRF3 nuclear localization in HEV infected HepG2/C3A cells. (B) Percentage of HEV-infected or ORF2+ cells with nuclear IRF3 translocation (ORF2+ TrIRF3+). (C) Percentage of ORF2 nuclear translocation (TrORF2+) among TrIRF3+ cells.

The following is now inserted in the result section (Lines 429-443): Next, to further investigate the effect of ORF2 nuclear localization on immune signaling, we carried out single-cell imaging flow cytometry experiments in HepG2/C3A cells (Figure 6a), which had more robustly upregulated ISGs upon HEV infection independently of pDCs (Figure 2a; left panel). Single-cell imaging flow cytometry allowed us to distinguish infected cells with (trORF2+) or without ORF2 nuclear translocation (trORF2-). We then examined these cells for IRF3 nuclear translocation, which is known to upregulate the expression of several cytokines, including IFN-I, IFN-III and CXCL10 (Brownell et al., 2014), as well as surface molecules. Secretion of chemokines can lead to the recruitment of certain immune cell subsets. For instance, CXCL10 secretion triggers pDC recruitment (Di Domizio et al., 2020).

We observed that around 10% of trORF2+ cells displayed IRF3 nuclear translocation (TrIRF3+) (Figure 6b). Around 80% of TrIRF3+ cells exhibited an ORF2 nuclear signal (Figure 6c). These results revealed that the immune status of IRF3 in HepG2/C3A infected cells can differ depending on the nuclear translocation of HEV ORF2, further validating the impact of ORF2 nuclear localization on the immune status of the cells.

The following is now inserted in the text in discussion section (Line 570-578): We also investigated whether ORF2 nuclear localization could be involved in the immune state of the infected cells. We showed that IRF3 nuclear translocation tends to occur more in HEV-

replicating cells with nuclear ORF2 than in HEV-replicating cells without nuclear ORF2. There could be several possible reasons for this activation of immune signaling, i.e., intrusion of nuclear membrane, sensing of the viral ORF2 in the nucleus, or sensing of accidentally shuttled HEV RNA in the nucleus. It would be interesting to investigate this further. The activation of IRF3 signaling could, in turn, aid pDCs in viral sensing via cytokines and upregulation of other effector molecules.

3. In the PLC3-pDC co-culture experiment (Fig. 2b), there is already an induction of IFN-λ1 (Interferon Lambda 1) in the uninfected PLC3-pDC co-culture (right panel, Fig. 2b). An explanation for the IFN-λ1 (Interferon Lambda 1) expression in the uninfected state would be helpful.

Please note that **Figure 2a-b** have been revised to include more datapoints. In line with the comments provided by this reviewer, in some independent experiments, we observed higher level of IFN-λ1 (Interferon Lambda 1) in the uninfected PLC3 cell-pDC co-culture. This is likely due to the steady-state level of IFN-λ1 in pDCs, which can vary across donors and between experimental conditions, due to cellular stress induced by slight differences in cell culture conditions. We would like to emphasize that the dataset of the conditions ‘No pDC’ *versus* ‘pDC co-culture’ are hard to compare since steady-state levels of gene expression are different in each cell type, especially for immune cells *versus* cell lines. Thus, the gene expression in coculture of pDCs and Control cells is affected by the basal levels in both pDCs and the cell line. The direct comparison between ‘No pDC’ *versus* ‘pDC co-culture’ dataset should be done with caution as the two datasets use separate housekeeping gene reference values for analysis. We have tried to limit direct comparisons among the datasets ‘No pDC’ *versus* ‘pDC co-culture’ by separating them in different panels.

We have now added the following statement in the legend of figure X (Line 768):

No pDC and pDC co-culture conditions have been separated into left and right panels as steady state levels of gene expression are different in these two datasets and therefore cross-comparisons between the two panels must be avoided.

Despite this additional caution for the comparison and analysis of the results, we observed that IFN-λ1 is significantly induced in pDC co-cultured with HEV-replicating cells *versus* control cells. Please find below a table with the round-up raw data that was plotted on Figure 2b.

Experiment number	Control cells	HEV cells	Controls +pDCs cells	HEV +pDCs cells
1	71	41	211	200 000
2	0	260	3	22 000
3	0	70	3620	741 000
4	466	43	na	na
5	170	63	202	108 929

6	116	795	na	na
7	252	333	836	1440
8			83	1010

Additional comments:

1) Authors checked the expression of two ISGs- MXA, ISG15 in Fig. 1a-c, 2a-b. Were the expressions of other ISGs, such as members of OAS family (OAS1, OAS2 etc.), IFITM family or any other ISGs checked? This may be helpful, since in the Fig. 2c there is IFN α production in pDC-infected PLC3 co-culture, but the ISGs (MXA, ISG15) are not upregulated significantly in Fig. 2b.

In accordance with the Reviewer's suggestions, we quantified the mRNA levels of OAS2, another ISG, in PLC3 cells and HepG2/C3A cells replicating HEV, with or without pDCs (**Novel Figure 2 below, data was added to Figure 2a-b**). Without pDCs, we noted a 4-log induction of OAS2 expression in HEV-replicating HepG2/C3A. However, the low number of repeats render statistical analysis inappropriate for this subset of experimental conditions (**Figure 2a-b, left panel**). With pDCs, OAS2 was significantly upregulated upon the pDC co-culture of HEV replicating HepG2/C3A *versus* control (**Figure 2a, right panel**). However, this difference was not observed for PLC3 cells (**Figure 2b, right panel**). OAS2 expression followed a trend similar to *ISG15* and *IFN- λ 1* expression, which were induced by HEV-replicating HepG2/C3A cells, independently of the presence of pDCs. In contrast, the expression of these genes remained unchanged in HEV-replicating PLC3 cells compared to control cells and even in the presence of pDCs. Overall, these results further highlight the differences in immune signaling between HepG2/C3A and PLC3 cells in response to HEV infection.

Novel Figure 2. OAS2 mRNA abundance in HEV-replicating cells, in the presence or absence of pDCs. (A-B) Quantification of the transcript expression levels of OAS2 in HepG2/C3A (A) and PLC3 (B) cells in the absence (left panels) or in co-culture with pDCs for 18 hours (right panels), as indicated.

We have now added the following statements in the result section:

(Line 213): *OAS2, which is known to activate RNase L antiviral activity, was also induced in HepG2/C3A (albeit not significantly).*

(Line 233-236): *We observed upregulated expression of all these effectors, and among these, MXA, ISG15, IFN- λ 1 and OAS2 were significantly upregulated when pDCs were co-cultured with HEV-replicating HepG2/C3A cells, as compared to basal levels with uninfected cells (Fig. 2a-b, right panels).*

2) In the HepG2/C3A-pDC co-culture experiment (Fig. 2a), there is not much difference in IFN- λ 1 (Interferon Lambda 1) level in the infected HepG2/C3A-pDC co-culture (right panel, Fig. 2a) in comparison to infected HepG2/C3A alone (left panel, Fig. 2b), and also this outcome is different from that in the PLC3 experiment (Fig. 2b). Further clarification would help to support the conclusion regarding the IFN- λ 1 (Interferon Lambda 1) upregulation in HEV infected cells-pDC co-culture.

As explained above, in response to major comment 3, we would like to emphasize that the 2 sets of qPCR data (no pDC *versus* pDC co-culture) should be considered separately (**revised Figure 2a-b**). In addition, to improve the reliability of the data, we have repeated the experiments to measure ISG response in the presence and absence of pDC for the two cell types (**revised Figure 2a-b**). In accordance with the previous results, we showed that IFN- λ 1 mRNA levels were enhanced in HEV-replicating HepG2/C3A cells and even more upon co-culture of pDCs.

We have performed ELISA analyses at the protein level for type III IFNs (IFN- λ 1 [IL-29], IFN- λ 2 [IL-28A] and IFN- λ 3 [IL-28B]) by ELISA (**Figure 2c-d**). These novel data showed that type III IFNs were significantly secreted upon 48h of co-culture of pDCs with HEV-replicating HepG2/C3A cells, as compared co-culture to control cells. This was not the case when PLC3 cells were co-cultured with pDCs, wherein very little or no IFN-III were secreted.

Figure 2. (C-D) pDCs were cocultured with HEV-electroporated PLC3 or HepG2/C3A cells for 18 hours and 48 hours. Quantification of IL-29/28B in supernatants of pDCs co-cultured with HEV-infected HepG2/C3A **(C)** and PLC3 cells **(D)**.

We have now added the following paragraph in the result section (Line 242):

We tested if this was true at the protein level by ELISA for lambda IFNs or type III IFNs (IFN- λ 1 [IL-29], IFN- λ 2 [IL-28A] and IFN- λ 3 [IL-28B]). We observed that IFN-III protein levels were significantly upregulated only at 48h post pDC co-culture with HEV-replicating HepG2/C3A cells, as compared to control cells co-cultured with pDCs (Figure 2c). Even though IFN- λ 1 mRNA levels were upregulated upon co-culture of pDCs with HEV-replicating PLC3 cells as early as 18 hours of coculture (Figure 2b; right panel); only a small amount of secreted IFN-III was detectable at the protein level 48 hours post co-culture (Figure 2d). This is possibly because of the lesser sensitivity of the ELISA over RT-qPCR analyses (i.e., lower dynamic range). Further analyses with more sensitive/advanced approaches for IFN-III detection will be needed. Therefore, IFN-III is more robustly upregulated in pDC coculture of HEV-replicating HepG2/C3A cells.

3) The authors show that in the pDC-PLC3 co-culture system, IFN α was induced at 18h (Fig. 2c-2e), but the viral replication was not decreased in PLC3 cells (Fig. 2g). But, the HepG2/C3A-pDC co-culture has reduced viral replication at 18h (Fig. 2f). An explanation for the difference in the observation in two different cell lines at the same timepoints would strengthen the antiviral role of pDCs on HEV infected cells.

We observed more efficient or faster viral control in HepG2/C3A cells compared to PLC3 cells. This was likely due to a **significant upregulation of ISGs** (*ISG15, IFN- λ 1, and OAS2*) - even in absence of pDCs - in HEV-replicating HepG2/C3A cells *versus* an inexistent one in HEV-replicating PLC3 cells. We hypothesize that the pre-existing ISG response in HEV-replicating HepG2/C3A, produced in the absence of pDCs, potentiates a more effective viral control in the presence of pDCs. Therefore, the initial ISG expression in infected cells may synergize the antiviral response by pDCs.

We have now added the following paragraph in the discussion (Lines 509-516):

We observed more efficient or faster viral control in HepG2/C3A cells compared to PLC3 cells. This was likely due to a significant upregulation of ISGs (ISG15, IFN- λ 1, and OAS2) - even in absence of pDCs - in HEV-replicating HepG2/C3A cells versus an inexistent one in HEV-replicating PLC3 cells. We hypothesize that the pre-existing ISG response in HEV-replicating HepG2/C3A, produced in the absence of pDCs, potentiates a more effective viral control in the presence of pDCs. Therefore, the initial ISG expression in infected cells may synergize the antiviral response by pDCs.

4) The authors quantified the fold change in HEV infected PLC3+ cells in Fig. 2h. Was it performed by flow cytometry? It would be helpful to mention it in the figure legend. Also, if the said quantitation was done by flow cytometry, performing similar assay with HEPG2/C3A cells at 48h would provide the readers a better idea about the antiviral response across the cell lines at comparable timepoints.

Sorry for this missing information. We have now added the following clause in the legend, "Quantification of ORF2 and GFP expressing cells by flow cytometry".

We have compared the two cell types for control of viral replication at 18h post co-culture with pDCs. Based on these results, we thus conducted further analysis focusing on PLC3 cells to study control of viral spread at 48h post co-culture with pDCs, since we did not observe control of viral replication in these cells at 18h post co-culture with pDCs.

We have now added the following sentence in the result section (Line 289-291):

As we did not observe pDC-mediated viral control in HEV-replicating PLC3 cells at this early time point, we thus studied the effect of pDCs when co-cultured for 48 hours.

Minor comments:

1) Was it expected to observe the increased induction of IL6 (Fig. 1b) in HepG2/C3A cells (but not in other cell lines) after IFN- λ (Interferon Lambda) treatment?

IL-6 has been shown to be induced upon IFN- λ 1 pretreatment in human primary gingival keratinocytes (HGK) (<https://link.springer.com/article/10.1007/s10753-022-01624-1>). In another study, treatment of whole peripheral blood mononuclear cells (PBMCs) with IFN- λ 1 upregulated the expression of IL-6. IFN- λ 1 treatment of whole peripheral blood mononuclear cells (PBMCs) upregulated the expression of IL-6 (<https://onlinelibrary.wiley.com/doi/10.1155/2011/349575>). Since HepG2/C3A cells are strong responders of immune stimuli, it is not surprising that IL6 is induced in HepG2/C3A cells upon IFN- λ 1 treatment.

2) In Fig. 3e, for the WT cells, 4 datapoints are visible while in the legend it is mentioned n=5.

We thank you for your comment. We have now corrected the scale. This enables the fifth datapoint to be visible.

3) Typo: IRS661 in line 263, 699, Figure 2e.

This was corrected.

4) Typo: 200 μ l in line 579.

This was corrected.

5) Catalogue number for ELISA kit is missing (Line 584).

This was corrected.

6) It would be helpful if the color code for the imaging in Supplementary figure 2f is provided on the top of the images, as it is provided in other images.

This was corrected.

Significance

This article by Joshi et al. provides insight about the role of pDCs in controlling the HEV infection. However, the importance of pDC-infected cell contact mediated IFN-I secretion in antiviral response has been previously shown by the authors' group (Assil et al., 2019, Cell Host & Microbe) and others as well (E.g., Yun et al., 2021, Sci. Immunol.). The involvement of integrin mediated cell adhesion and TLR signaling in mediating this response was also shown. Though this manuscript does not advance the field of pDC biology or virology significantly, it does provide better understanding of the pDC antiviral response in the landscape of HEV infection. Although, it is out of the scope of this manuscript, elucidation of the mechanistic regulation how ORF2g/c controls the pDC-infected cell contact would be of great interest and significance. Overall, this study could be of interest to a general audience, especially to the virologists and researchers working in pDC biology.

We thank you for your constructive feedback

Reviewer #2

Evidence, reproducibility and clarity

Joshi et al. discovered that human pDC isolated from blood of healthy donors incubated in coculture with hepatitis E virus (HEV) infected hepatoma cell lines were triggered in a cell-cell contact-dependent manner to mount interferon-alpha responses. This conclusion was supported by experiments in a Transwell setting in which pDC were separated from infected cells by a permeable membrane and under which conditions pDC were not triggered to mount interferon-alpha responses. Treatment of the coculture system with an TLR7 inhibitor or with antibodies against the cell adhesion receptors ICAM-I and alphaLbeta2-integrin either entirely inhibited or reduced the induction of interferon-alpha responses, respectively. HepG2/C3A cells infected with different ORF2 mutants induced overall similar interferon-alpha responses in pDC. In contrast, PLC3 cells infected with an ORF2 variant that is not properly translocated into the nucleus did not induce interferon-alpha responses in pDC, whereas the other variants induced normal responses. This observation indicated that in the context of PLC3 cells the subcellular localization of ORF2 in the nucleus is critical to induce interferon-alpha responses in pDC. Finally, quantification of contact points between pDC and infected cells supported the

conclusion that enhanced cell-cell contact was necessary to efficiently induce interferon-alpha responses in pDC.

1) Overall, the study was carefully carried out and shows interesting results. It is appreciated that responses of human pDC isolated from blood of healthy donors were analyzed. Furthermore, the stimulation of pDC with infected hepatoma cell lines is interesting. Earlier studies showed that infected cells might be better pDC stimulators than free virus. Furthermore, in line 140 of the manuscript the authors correctly state that pDCs are resistant to virtually all viruses. Thus, they should have included experiments in which they stimulate pDC with free HEV. Comparative analysis of the data presumably would further highlight the relevance of pDC triggering by infected cells.

We agree with the reviewer that this a critical point. In **revised Figure 2e (blue bar)**, we showed that cell-free HEV barely activated pDCs as opposed to the robust response to HEV infected cells (pink bar). This is in accordance with the absence of pDC IFN α production in the coculture experiments performed in Transwell devices, in which pDCs are physically separated from infected cells.

We have emphasised this further in the text now (Line 260-264):

We also treated pDCs with cell-free HEV to verify if circulating virus particles can also activate pDCs in a cell-independent manner. pDCs exposed to the filtered supernatant [SN] collected from HEV-replicating cells very modestly secreted IFN α (Fig. 2e), suggesting that physical contact between pDCs and HEV-replicating cells is required for the robust pDC IFN-I secretion.

2) Despite interesting observations are presented regarding differences in the induction of pDC responses with PLC3 and HepG2/C3A cells infected with an ORF2 variant that is not properly translocated into the nucleus, no experiments were offered to explain this phenomenon. It is highly recommended to include additional experiments that support concepts either in one or the other direction as presented in the discussion.

Our results showed a reduction of pDC response when the 5R/5A mutant was expressed, as compared to the wild-type protein, both in PLC3 and HepG2/C3A cells. This reduction was more pronounced in PLC3 cells as compared to HepG2/C3 cells. Although this reduction was statistically significant in HepG2/C3A, our results uncovered that the intensity of ORF2i-mediated regulation of pDC response might be modulated by cell type-specific alterations upon HEV infection. Similar to our response to point 1, we proposed that this can result, at least in part, from cell type-dependent differences in inflammatory/innate immunity responsiveness among the two cell types. Notably, we observed that HEV infection differentially impacts the expression of *Tumor necrosis factor (TNF)*, a representative cytokine

of *NFKB*-induced signaling (**Figure 2a**). Interestingly, *TNF* is known to impact the expression of regulators of cell adhesion and/or immune cells recruitment, including *ICAM1* (<https://doi.org/10.3858/emm.2008.40.2.167>) and various inflammatory chemokines (<https://doi.org/10.1016/j.cytogfr.2014.07.016>).

We thus decided to analyze how 5R/5A mutation impacts *TNF* expression in the two cell lines (**Supplementary figure 3**). We found that 5R/5A mutant significantly reduced *TNF* induction in PLC3 cells but not in HepG2/C3A cells as compared to HEV WT. Our results thus show a cell type-specific regulation of *TNF* expression – potentially along with *NFKB*-induced signaling - depending on the nuclear translocation of ORF2, implying potential differential modulation of downstream effectors, including those involved in pDC adhesion/recruitment (e.g. ICAM), and hereby leading to weaker contacts between PLC3 cells and pDCs *versus* HepG2/C3A cells and pDCs in absence of ORF2 nuclear translocation.

The following is now inserted in the result section (Lines 385-397):

*Here, we showed that the pDCs exhibited reduced IFN α response to HepG2/C3A expressing the 5R/5A mutant, and a complete absence of this response in PLC3 cells. This is likely due to differences in immune signaling among the two cell lines with the 5R/5A mutant. We observed that HEV infection impacted, in a cell-type dependent manner, the expression of *TNF* (Figure 2a), a representative cytokine of *NFKB* signaling, known to modulate the expression of regulators of pDC adhesion and/or recruitment, including *ICAM1* (Kim et al., 2008) and various inflammatory chemokines (Sedger & McDermott, 2014). We thus decided to analyze the effect of 5R/5A mutation on *TNF* expression in these two cell lines (Supplementary figure 3a-b). We found that 5R/5A mutant significantly reduced *TNF* induction in PLC3 cells but not in HepG2/C3A cells as compared to HEV-WT, thus potentially explaining the cell type-specific regulation by ORF2i. Taken together, our results show that ORF2 protein expression and localization in producer cells modulate pDC response, with a magnitude that is cell-type dependent.*

The following is now inserted in the discussion section (Lines 562-570):

*The difference in pDC-mediated IFN-I response to WT and 5R/5A mutant is particularly interesting and was observed for both cell lines, even though at different magnitudes. The 5R/5A mutant, which is characterized by an absence of nuclear ORF2, displayed a reduced or abolished pDC response i.e., depending on the cell line. We then uncovered a cell line-specific induction of *TNF*, depending on ORF2 nuclear translocation. It indicates a possible differential modulation of downstream effector(s), including those involved in pDC cell adhesion/recruitment, and eventually leading to a weaker interferogenic synapse between PLC3 cells and pDCs, but not at play in HepG2/C3A cells. Further investigation will be needed to identify the effector(s) at play.*

Novel supplementary Figure 3. Abundance of *TNF* expression in HepG2/C3A and PLC3 cells upon HEV replication. Quantification of the transcript levels of *TNF* determined at 6 d.p.e. in HepG2/C3A (A) and PLC3 (B) cells. Bars represent copy number per µg total RNA; means ± SD; each dot represents one independent experiment (n=5 to 7). Statistical analysis was done using Wilcoxon rank sum test with continuity correction; *p* values as: * ≤0.05, ** ≤0.005 and *** ≤0.0005.

To further investigate the effect of ORF2 nuclear localization on immune signaling, we carried out single-cell imaging flow cytometry experiments in HepG2/C3A cells, which demonstrated more efficient upregulation of ISGs than in PLC3 cells, therefore also higher probability of IRF3 nuclear localization. This approach allowed to distinguish infected cells with (trORF2+) or without ORF2 nuclear translocation (trORF2-) (**Figure 6a**). We then examined these cells for IRF3 nuclear translocation, which is known to upregulate the expression of several cytokines, including IFN-I, IFN-III, and CXCL10 (doi: 10.1128/JVI.02007-13), as well as surface molecules. Secretion of chemokines can lead to the recruitment of certain immune cell subsets. For instance, CXCL10 secretion triggers pDC recruitment (<https://doi-org.proxy.insermbiblio.inist.fr/10.1038/s41590-020-0721-6>).

Novel figure 6a. Single-cell analysis of IRF3 nuclear localization in HEV infected HepG2/C3A cells. HepG2/C3A cells expressing IRF3-GFP electroporated with WT HEV RNA and fixed 2 d.p.e. after staining with viability stain (Zombie aqua), ORF2 (APC) and DNA (Hoechst). Single-cell images of viable ORF2+ cells were obtained and categorized according to the localization of IRF3 and ORF2. **(A)** Cells with translocation of ORF2 and IRF3 into the nucleus are represented by TrORF2+ and TrIRF3+. Cells without translocation of ORF2 and IRF3 into the nucleus are represented by TrORF2- and TrIRF3-.

We observed that around 10% of trORF2+ cells displayed IRF3 nuclear translocation (TrIRF3+) **(Figure 6b)**. Around 80% of TrIRF3+ cells exhibited an ORF2 nuclear signal **(Figure 6c)**. These results revealed that the immune status of IRF3 in HepG2/C3A infected cells can differ depending on the nuclear translocation of HEV ORF2, further validating the impact of ORF2 nuclear localization on the immune status of the cells.

Figure 6b and c. Single-cell analysis of IRF3 nuclear localization in HEV infected HepG2/C3A cells. (B) Percentage of HEV-infected or ORF2+ cells with nuclear IRF3 translocation (ORF2+ TrIRF3+). (C) Percentage of ORF2 nuclear translocation (TrORF2+) among TrIRF3+ cells.

The following is now inserted in the result section (Lines 429-443): Next, to further investigate the effect of ORF2 nuclear localization on immune signaling, we carried out single-cell imaging flow cytometry experiments in HepG2/C3A cells (Figure 6a), which had more robustly upregulated ISGs upon HEV infection independently of pDCs (Figure 2a; left panel). Single-cell imaging flow cytometry allowed us to distinguish infected cells with (trORF2+) or without ORF2 nuclear translocation (trORF2-). We then examined these cells for IRF3 nuclear translocation, which is known to upregulate the expression of several cytokines, including IFN-I, IFN-III and CXCL10 (Brownell et al., 2014), as well as surface molecules. Secretion of chemokines can lead to the recruitment of certain immune cell subsets. For instance, CXCL10 secretion triggers pDC recruitment (Di Domizio et al., 2020).

We observed that around 10% of trORF2+ cells displayed IRF3 nuclear translocation (TrIRF3+) (Figure 6b). Around 80% of TrIRF3+ cells exhibited an ORF2 nuclear signal (Figure 6c). These results revealed that the immune status of IRF3 in HepG2/C3A infected cells can differ depending on the nuclear translocation of HEV ORF2, further validating the impact of ORF2 nuclear localization on the immune status of the cells.

The following is now inserted in the text in discussion section (Line 570-578): We also investigated whether ORF2 nuclear localization could be involved in the immune state of the infected cells. We showed that IRF3 nuclear translocation tends to occur more in HEV-replicating cells with nuclear ORF2 than in HEV-replicating cells without nuclear ORF2. There could be several possible reasons for this activation of immune signaling, i.e., intrusion of nuclear membrane, sensing of the viral ORF2 in the nucleus, or sensing of accidentally

shuttled HEV RNA in the nucleus. It would be interesting to investigate this further. The activation of IRF3 signaling could, in turn, aid pDCs in viral sensing via cytokines and upregulation of other effector molecules.

3) The paper is written in an unnecessarily complicated way. The authors should try to arrange the manuscript in a manner that the readability is enhanced. In the end, the observations that have to be communicated are not very complicated. Showing only controls in Fig. 1 is a cumbersome start in the manuscript. The authors should consider moving such results in a supplementary Figure. The authors do not thoroughly describe the experiment in Fig. 2I in the results section, although it seems to be rather interesting. The authors should give more explanations in the results section how they quantified cell-to-cell HEV infection in presence or absence of pDC.

We thank this Reviewer for the suggestion. We have now moved **Figure 1d-e** to supplementary figure as recommended.

In addition, we have modified the explanation about Figure 3 in the results section (Line 295-303): *To further examine whether the pDC-mediated IFN response decreases the percentage of newly infected cells, we co-cultured a mixture of PLC3 cells, electroporated with HEV RNA and not expressing GFP (ORF2+/GFP- cells) along with uninfected cells positive for GFP (ORF2-/GFP+) in the presence or absence of pDCs for 48 hours. The impact of pDC response on HEV spread was assessed by flow cytometry to quantify the percentage of newly infected cells, i.e., GFP+ cells that became ORF2+ due to viral spread (ORF2+/GFP+) in the presence versus absence of pDCs (Fig. 3d-e). Infected cells were distinguished from pDCs on the basis of size (FSC-SSC gating), and then the infected cell type (PLC3 cells) was gated for expression of ORF2 and/or GFP (Fig. 3d).*

To further emphasize the distinct questions addressed, better clarify the rational and flow of the manuscript, we have now changed the set of Figures, as followed:

The results obtained on the expression changes in pDCs, in response to HEV-infected cells i.e., ISG and IFN-I/III protein expression (by ELISA and RT-qPCR) are included in **Figure 2**. The results demonstrating the control of viral infection/spread by IFN producing pDCs are now a separate **Figure 3**.

We have made the following additions and changes to simplify the text and enhance readability:

Rephrased Line 37: *Previous studies have shown that IFN-mediated antiviral responses against hepatitis E virus (HEV) are suppressed and defeated by viral escape mechanisms at play in infected hepatocytes.*

Rephrased Line 61: *HEV infection is often asymptomatic and resolves on its own in case of healthy subjects. However, severe cases have mainly been reported in pregnant women,*

while chronic infections are more common in immunocompromised patients (Lhomme et al., 2020). This makes host immunity a crucial factor in influencing the outcome of the disease.

Rephrased Line 77: *HEV infection has been recognized as a burgeoning issue in industrialized countries due to its chronicity in immunocompromised gt3-infected patients, the transmission of HEV through blood transfusion, a growing number of diagnosed HEV cases, and complications in patients with pre-existing liver disease (Sayed et al., 2017).*

Rephrased Line 352: *This attested comparable levels of HEV RNA, which is a prerequisite for testing the impact of this mutant panel on the response of co-cultured pDC.*

Rephrased Line 370: *Since comparable HEV RNA levels were quantified in NES versus 5R/5A mutant (Fig. 4d), the nuclear localization of ORF2, which is abrogated for the 5R/5A mutant, may affect pDC response against HEV.*

Minor comments

1) In the legend of Fig. 1D and E it should be clarified what "Cont cells" and "HEV cells" means. I assume it is the mock control and HEV infected cells. It should be mentioned in the figure caption with which multiplicity of infections the cells were treated.

Figures 1D and 1E were moved to supplementary Figures 1A and 1B and the legend was modified to: *"Quantification of the replication levels of HEV at 6 days post electroporation with 10 μ g RNA [HEV cells] or mock electroporation without RNA [Cont cells] (as described in Material and method section) by RT-qPCR detection of HEV RNA (amplicon/primers designed in ORF2) in HepG2/C3A (A) and PLC3 cells (B)"*

We also modified the legend of main Figure 2A-B to: *"Quantification of the transcript expression levels of representatives of IFN-I/ λ -signaling (i.e., MxA, ISG15, OAS2 and IFN λ 1) and NF-KB-related pathway (i.e., TNF and IL-6) determined at 6 d.p.e. of HepG2/C3A (A) and PLC3 (B) cells that were electroporated with 10 μ g RNA [HEV cells] or mock electroporated without RNA [Cont cells], in the absence (left panels) or in co-culture with pDCs for 18 hours (right panels), as indicated."*

2) For reasons of consistency, in Fig. 2 controls should be identified also by "Cont cells" and not by "cont cell" (in A and B) and "cont cells" (in C and D).

This was corrected.

3) In Fig. 2H, above "Fold-change in ORF2+ cells" there is indicated PLC3 in small letters. Should this be better moved above the panel, as done in F and G?

This was corrected in revised Figure 3c.

4) The schematic depiction of different cell types in Fig. 2H and J is not very helpful, if not explained in the figure caption.

The schematic depiction (**now revised Figure 3c**) is now described: “Quantification of ORF2 expressing cells in the presence or absence of pDCs by flow cytometry” (line 732).

The schematic depiction in Fig. 2J (**now Figure 3e**) was moved to 2I (**now Figure 3d**) and described: “HEV-infected PLC3 cells (GFP- ORF2+) and uninfected cells (GFP+ ORF2-) were co-cultured in the presence and absence of pDCs for 48 hours” (line 734).

5) In Fig. 3B the labels shown on the left side of A should be repeated.

This was corrected.

6) Why above the second row in A and B it is indicated "ORF2-AF488", but everywhere else in the manuscript it is referred to ORF2. If the AF488 addition is necessary (presumably it refers to an Alexa Fluor 488 conjugate of the antibody used to detect ORF2), this should be introduced either in the Materials and Methods section or in the figure caption.

In accordance with this comment, we changed ORF2-AF488 to ORF2 in **Figure 4a-b**.

We also modified the Materials and Methods section (line 685): *After another wash, cells were incubated with a secondary antibody, i.e., Goat anti-Mouse IgG2b Cross-Adsorbed Secondary Antibody Alexa Fluor 488 (Thermo Fisher Scientific Catalog #A-21141) for 2 hours.*

We have also added details in the figure caption (line 818): *1E6 antibody, followed by targeting secondary antibody with AlexaFluor488.*

Significance

The study is of high relevance. The presented phenomena are clearly described. Mechanistic aspects are not fully covered, yet. This study will be interesting for a relatively broad audience of virologists and immunologists. Currently, the manuscript is written in a manner that it is relatively difficult to read. This should be improved to reach the relatively divergent audience.

We thank this Reviewer for her/his positive appreciation of our study.

February 24, 2025

RE: Life Science Alliance Manuscript #LSA-2025-03256

Dr. Marlene - Dreux
Centre International de Recherche en Infectiologie
50 avenue Tony Garnier
Lyon, Other / Non-US 69007
France

Dear Dr. Dreux,

Thank you for submitting your revised manuscript entitled "Hepatitis E Virus-induced antiviral response by plasmacytoid dendritic cells is modulated by the ORF2 protein". We would be happy to publish your paper in Life Science Alliance pending final revisions necessary to meet our formatting guidelines.

- please address Reviewer 1's remaining comments
- please be sure that the authorship listing and order is correct
- please upload your main manuscript text as an editable doc file
- please upload your main and supplementary figures as single files
- please add a Running Title to our system
- please add a Summary Blurb/Alternate Abstract and a Category to our system
- please add the X and Bluesky handles of your host institute/organization as well as your own or/and one of the authors in our system
- please add your main, supplementary figure, and table legends to the main manuscript text after the references section
- please add an Author Contributions section to your main manuscript text
- please add a Conflict of Interest statement to your main manuscript text
- please use the [10 author names, et al.] format in your references (i.e. limit the author names to the first 10)
- please upload your Table in editable .doc or excel format
- please add a callout for Figure S1E to your main manuscript text

FIGURE CHECK

- please add scale bars to Figure S1C

A. FINAL FILES:

-- Summary blurb (enter in submission system): A short text summarizing in a single sentence the study (max. 200 characters including spaces). This text is used in conjunction with the titles of papers, hence should be informative and complementary to

the title. It should describe the context and significance of the findings for a general readership; it should be written in the present tense and refer to the work in the third person. Author names should not be mentioned.

B. MANUSCRIPT ORGANIZATION AND FORMATTING:

Sincerely,

Reviewer #1 (Comments to the Authors (Required)):

The manuscript by Joshi et al. investigates the role of plasmacytoid dendritic cells (pDCs), a specialized interferon (IFN) producing cell type, in regulating the Hepatitis E virus (HEV) infection. By utilizing a combination of imaging, flow cytometry and other techniques, the authors report that the cell-cell contacts between the pDCs and HEV infected cells induce the pDCs to secrete IFN. This interaction is mediated by cell adhesion molecules and is dependent on TLR7 signaling. The authors then demonstrated that the IFN produced by pDCs controlled the viral spread. Further, using several mutant forms of ORF2 protein, the authors elucidated the importance of the glycosylation pattern and subcellular localization of different forms of HEV ORF2 protein in triggering the immune response. Finally, by imaging flow cytometry, the authors correlated the nuclear localization of HEV ORF2 and the immune response. Overall, this study provided insights in the pDC mediated IFN response against HEV.

Comments:

1) Generally, IFN- 1 and IL-6 are not considered as the canonical ISGs. Therefore, rephrasing the lines 213 and 483 would be helpful.

2) Was the supplementary figure 1e supposed to be referred in line 319?

3) In supplementary figure 4C legend, n=3 is mentioned, but there are more data points in the figure. Are they from different imaging fields in total 3 independent images or each dot represents an independent image as mentioned in the legend (in that case n would change)? Also, in figure 5B and 5C legend, n=4 was mentioned but the datapoints are more than that.

The authors have addressed all the previous comments.

Significance:

This manuscript by Joshi et al. provides insight about the role of pDCs in controlling the HEV infection. However, the importance of pDC-infected cell contact mediated IFN-I secretion in antiviral response was previously shown by the author's group (Assil et al., 2019, Cell Host & Microbe) and others as well (E.g., Yun et al., 2021, Sci. Immunol.). The involvement of integrin mediated cell adhesion and TLR signaling in mediating this response was also reported. Though this manuscript does not advance the field of pDC biology or virology significantly, it does provide better understanding of the pDC antiviral response in the landscape of HEV infection. Although, it is out of the scope of this manuscript, future exploration of the mechanistic regulation how

ORF2g/c controls the pDC-infected cell contact would be of great interest and significance. Overall, this study would be of interest to a general audience, and especially to the virologists and immunologists.

Reviewer #2 (Comments to the Authors (Required)):

The authors carefully addressed all reviewers' comments and implemented corresponding changes in the manuscript. This way the overall quality and readability of the manuscript was significantly enhanced.

Dear reviewers and editor,

We would like to express our sincere gratitude to you for the time and effort you invested in reviewing our manuscript. Your insightful comments and constructive feedback have greatly enhanced the quality of our work. We truly appreciate your valuable contributions, which have helped us refine our article. Here are the responses to your remaining comments.

Reviewer #1 (Comments to the Authors (Required):

The manuscript by Joshi *et al.* investigates the role of plasmacytoid dendritic cells (pDCs), a specialized interferon (IFN) producing cell type, in regulating the Hepatitis E virus (HEV) infection. By utilizing a combination of imaging, flow cytometry and other techniques, the authors report that the cell-cell contacts between the pDCs and HEV infected cells induce the pDCs to secrete IFN. This interaction is mediated by cell adhesion molecules and is dependent on TLR7 signaling. The authors then demonstrated that the IFN produced by pDCs controlled the viral spread. Further, using several mutant forms of ORF2 protein, the authors elucidated the importance of the glycosylation pattern and subcellular localization of different forms of HEV ORF2 protein in triggering the immune response. Finally, by imaging flow cytometry, the authors correlated the nuclear localization of HEV ORF2 and the immune response. Overall, this study provided insights in the pDC mediated IFN response against HEV.

Comments:

1) Generally, IFN- λ 1 and IL-6 are not considered as the canonical ISGs. Therefore, rephrasing the lines 213 and 483 would be helpful.

We have now modified lines 236-238 (previously 213), as follow:

Next, we looked at expression levels of three ISGs (*MXA*, *ISG15* and *OAS2*) and three cytokines (*TNF*, *IL-6* and *IFN- λ 1*) in HEV-replicating HepG2/C3A and PLC3 cells (**Fig. 2a-b**). We found a significant increase in mRNA levels of *ISG15*, *IL-6*, *IFN- λ 1* in HepG2/C3A cells at 6 d.p.e. (**Fig. 2a**; left panel).

We have now modified lines 508 (previously 483), as follow:

Upon co-culture of HEV-replicating HepG2/C3A cells with pDCs, *the 3 ISGs MXA, ISG15 and OAS2, as well as IFN- λ 1*, were robustly upregulated.

2) Was the supplementary figure 1e supposed to be referred in line 342?

We have now modified lines 319, as follow:

To address whether these distinct ORF2 forms impact the host immune responses in infected cells, and consequently and/or additionally the sensing of infected cells by pDCs, we tested a series of ORF2 protein mutants of the p6 HEV strain (WT) (**Supplementary Fig. 1e**).

3) In supplementary figure 4C legend, n=3 is mentioned, but there are more data points in the figure. Are they from different imaging fields in total 3 independent

images or each dot represents an independent image as mentioned in the legend (in that case n would change)?

Each dot represents a single image (and not different imaging fields) from three distinct experiments. We have now modified the legend of Supplementary Figure 4c for better understanding:

'Violin plots present median and each dot for independent image ($n \geq 6$) from 3 distinct experiments'

Also, in figure 5B and 5C legend, n=4 was mentioned but the datapoints are more than that.

We have now modified the legend of Figure 5b-c for better understanding:

(B) Frequency of HEV-replicating (ORF2⁺) PLC3 cells detected among the cells defined as non-pDC (CTV⁻/CMTPX⁺); bars present means \pm SD and each dot for independent image ($n \geq 8$) from 4 distinct experiments. **(C)** Contact/proximity of PLC3 infected cells and pDCs with detection of CTV/pDCs and CMTPX⁺ ORF2⁺/infected cells, as reference; bars present means \pm SD and each dot for independent image ($n \geq 8$) from 4 distinct experiments.

The authors have addressed all the previous comments.

Significance:

This manuscript by Joshi et al. provides insight about the role of pDCs in controlling the HEV infection. However, the importance of pDC-infected cell contact mediated IFN-I secretion in antiviral response was previously shown by the author's group (Assil et al., 2019, Cell Host & Microbe) and others as well (E.g., Yun et al., 2021, Sci. Immunol.). The involvement of integrin mediated cell adhesion and TLR signaling in mediating this response was also reported. Though this manuscript does not advance the field of pDC biology or virology significantly, it does provide better understanding of the pDC antiviral response in the landscape of HEV infection. Although, it is out of the scope of this manuscript, future exploration of the mechanistic regulation how ORF2g/c controls the pDC-infected cell contact would be of great interest and significance. Overall, this study would be of interest to a general audience, and especially to the virologists and immunologists.

Reviewer #2 (Comments to the Authors (Required)):

The authors carefully addressed all reviewers' comments and implemented corresponding changes in the manuscript. This way the overall quality and readability of the manuscript was significantly enhanced.

March 14, 2025

RE: Life Science Alliance Manuscript #LSA-2025-03256R

Dr. Garima Joshi
Centre International de Recherche en Infectiologie
France

Dear Dr. Joshi,

Thank you for submitting your Research Article entitled "Plasmacytoid dendritic cell sensing of Hepatitis E Virus is shaped by both viral and host factors". It is a pleasure to let you know that your manuscript is now accepted for publication in Life Science Alliance. Congratulations on this interesting work.

DISTRIBUTION OF MATERIALS:

Again, congratulations on a very nice paper. I hope you found the review process to be constructive and are pleased with how the manuscript was handled editorially. We look forward to future exciting submissions from your lab.

Sincerely,
